

# Organic carbon characteristics in yedoma and thermokarst deposits on Baldwin Peninsula, West-Alaska

Loeka L. Jongejans[1], Jens Strauss[1], Josefine Lenz[1,2], Francien Peterse[3], Kai Mangelsdorf[4], Matthias Fuchs[1,5] and Guido Grosse[1,5]

[1]Alfred Wegener Institute Helmholtz Centre for Polar and Marine Research, Periglacial Research Section, Potsdam, Germany
[2]University of Alaska Fairbanks, Institute of Northern Engineering, Fairbanks, USA
[3]Utrecht University, Department of Earth Sciences, Utrecht, Netherlands
[4]Helmholtz Centre Potsdam - German Research Centre for Geosciences, Germany
[5]University of Potsdam, Institute of Earth and Environmental Sciences, Potsdam, Germany

*Correspondence to:* Loeka L. Jongejans (loeka.jongejans@awi.de)

**Abstract.** As Arctic warming continues and permafrost thaws, more soil and sedimentary organic carbon (OC) will be decomposed in northern high latitudes. Still, uncertainties remain in the quantity and quality of OC stored in different deposit types of permafrost landscapes. This study presents OC data from deep permafrost and lake deposits on the Baldwin

Peninsula which is located in the southern portion of the continuous permafrost zone in West-Alaska. Sediment samples from yedoma and drained thermokarst lake basin (DTLB) deposits as well as thermokarst lake sediments were analyzed for cryostratigraphical and biogeochemical parameters and their lipid biomarker composition to identify the size and quality of belowground OC pools in ice-rich permafrost on Baldwin Peninsula. We provide the first detailed characterization of yedoma deposits on Baldwin Peninsula. We show that three quarters of soil organic carbon in the frozen deposits of the

study region (total of 68 Mt) is stored in DTLB deposits (52 Mt) and one quarter in the frozen yedoma deposits (16 Mt). The lake sediments contain a relatively small OC pool (4 Mt), but have the highest volumetric OC content (93 kg m$^{-3}$) compared to the DTLB (35 kg m$^{-3}$) and yedoma deposits (8 kg m$^{-3}$), largely due to differences in the ground ice content. The biomarker analysis indicates that the OC in both yedoma and DTLB deposits is mainly of terrestrial origin. Nevertheless, the relatively high carbon preference index of plant leaf waxes in combination with a lack of degradation trend with depth in the yedoma

deposits indicates that OC stored in yedoma is less degraded than that stored in DTLB deposits. This suggests that OC in yedoma has a higher potential for decomposition upon thaw, despite the relatively small size of this pool. These findings highlight the importance of molecular OC analysis for determining the potential future greenhouse gas emissions from thawing permafrost, especially because this area close to the discontinuous permafrost boundary is projected to thaw substantially within the 21$^{st}$ century.



# 1 Introduction

The Arctic region is warming twice as fast as the global mean (Overland et al., 2017). Ice-rich permafrost soils are particularly vulnerable to climate warming and susceptible to large-scale thermokarst processes. Thermokarst is the subsidence of ground resulting from the thawing of ice-rich permafrost (Grosse et al., 2013; Günther et al., 2013).

Thermokarst lake development often starts with the coalescence of polygonal ponds after the degradation of ice wedges (Grosse et al., 2013). This is followed by the formation of a body of unfrozen ground underneath the lake (i.e. talik) and subsequently, both the lake and talik grow and deepen. Thermokarst lake development can cease by drainage through lateral outflows formed by thermal erosion of ice wedge networks, by tapping of lakes due to coastal erosion, by vertical outflow through open taliks in regions with thin permafrost, or by infilling with sediment or encroaching vegetation (Burn and Smith,

1990; Jones and Arp, 2015; Lenz et al., 2016c). After lake loss, the remaining basins largely become vegetated and eventually permafrost can reform (Jones et al., 2012). Due to the different stages of lake development, thermokarst landscapes are highly dynamic and form complex patterns of landscape units (Jones et al., 2012; Lenz et al., 2016a).

Late Pleistocene, ice-rich syngenetic permafrost, known as yedoma, is especially prone to rapid and deep thaw processes (Schirrmeister et al., 2013). These deposits cover large regions of Siberia and Alaska (Kanevskiy et al., 2011;

Schirrmeister et al., 2013; Strauss et al., 2017) and can reach a thickness of up to 50 m (Sher, 1997; Shur et al., 2012). Yedoma contains large syngenetic ice-wedges and can have a ground ice content of up to 80 vol%, thus yedoma deposits are highly vulnerable to thermokarst processes (Kanevskiy et al., 2016; Ulrich et al., 2014).

Permafrost landscapes store large quantities of OC (Hugelius et al., 2014; Strauss et al., 2013) as the generally low decomposition rates, due to low soil temperatures and poor drainage, inhibit decomposition of organic matter

(Davidson and Janssens, 2006). Permafrost thaw and talik formation, however, allow microbial decomposition of the previously freeze-locked OC, resulting in increased greenhouse gas emissions (in particular carbon dioxide and methane) into the atmosphere and enhancing the initial warming (Koven et al., 2015; Schuur et al., 2015; Strauss et al., 2017). Rapid and large-scale permafrost degradation of ice-rich permafrost, such as yedoma deposits, may constitute a positive feedback to atmospheric warming.

Due to the freezing conditions during and after accumulation, the OC stored in yedoma deposits is highly decomposable (Knoblauch et al., 2013; Schädel et al., 2014; Strauss et al., 2017). Yedoma deposits contain on average less OC than DTLB deposits (e.g. Strauss et al., 2013), however higher respiration rates from yedoma deposits have been observed compared to DTLB deposits (Dutta et al., 2006; Lee et al., 2012; Zimov et al., 2006). Moreover, previous studies found that the state of OC decomposition is more dependent on OC properties, rather than on OC age (Knoblauch et al., 2013; Stapel et al., 2016).

It is therefore crucial to characterize the quantity and quality of OC pools in both undisturbed as well as disturbed ice-rich permafrost landscapes, so that the potential contribution to future greenhouse gas release following permafrost degradation can be better constrained for each of the different landscapes.



This study aims to characterize the belowground OC stored in thermokarst affected areas by analyzing sediment samples from yedoma deposits, drained thermokarst lake basin deposits and thermokarst lake sediments. Two main questions are addressed: 1) how much OC is stored and 2) what is the quality of this OC? In this study, we particularly refer to belowground OC and the quality of OC refers to the decomposability of the material, respectively the vulnerability to future

decomposition. In our analysis, we focus on cryostratigraphical (bulk density, absolute ice content) and biogeochemical parameters (total organic carbon, total nitrogen, C/N ratio and stable carbon isotopes) to estimate the OC pool size. In addition, lipid plant and soil specific biomarkers (*n*-alkanes and brGDGTs, respectively) were used to determine the molecular composition of each OC pool and its quality.

## 2 Material and methods

### 2.1 Study area

The study sites are located on the western coast of the Baldwin Peninsula, which is surrounded by the Kotzebue Sound in Northwest Alaska (66°40' N, 162°15' W) (Figure 1). The Baldwin Peninsula is part of the former landmass Beringia that remained unglaciated during the Late Pleistocene (Hopkins, 1982). The peninsula is located in southern portion of the continuous permafrost zone and therefore close to the discontinuous permafrost boundary (Jorgenson et al., 2008).

According to geological maps, the peninsula is largely composed of a sequence of marine, fluvial and glaciogenic sediments which are well exposed along coastal bluffs and in some regions covered by loess-like deposits (Hopkins et al., 1961; Huston et al., 1990; Pushkar et al., 1999). A field campaign in summer 2016 revealed several yedoma exposures in active retrogressive thaw slumps on the western coast of the peninsula that consisted of fine-grained, ice-rich permafrost deposits penetrated by large syngenetic ice wedges. These yedoma exposures were located in upland remnants, a setting which is

typical for yedoma hills found in northeast Siberia (Schirrmeister et al., 2013). A large portion of the peninsula has been affected by severe permafrost degradation and multiple thermokarst lake basin generations visible in satellite imagery (Figure 1). The present-day climate is subarctic with an average annual precipitation of 280 mm and a mean annual air temperature of -5.2°C (US Climate Data, 2017).

### 2.3 Field work

Sampling of yedoma and DTLB exposures and thermokarst lake sediments was carried out on the western coast of the Baldwin Peninsula in August 2016. Photographs of the exposures and the lake core are shown in Supplement 2.1.

A yedoma coastal bluff (BAL16-B2) of 16 m thickness was sampled. This bluff is characterized by sediments containing large ice wedges, ice bands and ice lenses, which are overlying a separate unit of ice-rich silty sediments. 25 representative samples of the yedoma deposits were collected using a handheld power drill (ø 57 mm) at five different sampling locations

along the exposure. Three additional samples were taken from the separate depositional unit underlying the yedoma which was not as ice-rich and did not contain in-situ formed ice wedges. Exposures of DTLB deposits ranging in thickness from 3



to 8 m were sampled from three exposed profiles (BAL16-B3, BAL16-B4 and BAL16-B5). 31 samples were collected from the 8-m-high DTLB exposure BAL16-B4 consisting of mostly laminated sediments that contained various cryotextural and organic features (e.g. ice inclusions, lenses, organic matter inclusions). The DTLB exposures BAL16-B3 (9 samples) and BAL16-B5 (4 samples) were sampled to generate a broader sample base for the estimates of the DTLB OC pool (the results

of the two DTLB additional profiles BAL16-B3 and BAL16-B5 are shown by depth in Supplement 2.2). All samples from the yedoma and DTLB exposures were kept frozen until laboratory analysis at Alfred Wegener Institute (AWI), Helmholtz Centre for Polar and Marine Research, in Potsdam.

In addition, a 34-cm-long sediment core (BAL16-UPL1-L1) was retrieved from a shallow thermokarst lake (1.35 m deep at coring site) with a N-W extent, a length of about 1,400 m and a width of 800 m. Unfrozen, slightly layered, clayey silty

lake sediments were retrieved in a PVC tube (ø 60 mm) using a piston corer and were kept cool until sub-sampling (9 samples) and laboratory analysis.

## 2.4 Bulk physical properties

In the laboratory, the sediment samples were freeze-dried. The absolute ice content was derived from the weight difference of the frozen and dry samples after Eq. (1) and is expressed in weight percentage (wt%).

$$Absolute\ ice\ content\ [wt\%] = \frac{wet\ weight - dry\ weight}{wet\ weight} * 100 \tag{1}$$

For the frozen samples, the bulk density $BD_1$ was calculated following Eq. (2).

$$BD_1[10^3\ kg\ m^{-3}] = (\varphi - 1) * -\rho_s \tag{2}$$

where $\varphi$ is porosity (fraction) and $\rho_s$ is the mineral density in $[10^3\ kg\ m^{-3}]$. The porosity is derived from the volumetric ratio of ice and dry sample whereby a constant ice density was assumed of $0.91*10^3\ kg\ m^{-3}$ (at 0°C) (Lide, 1999). For the mineral

density, the density of quartz of $2.65*10^3\ kg\ m^{-3}$ was taken (Rowell, 1994).

For the unfrozen thermokarst lake sediments, $BD_2$ was calculated following Eq. (3). The volume of the sediment was corrected to account for the compression of the sediment during sampling. The volume of the excess water in the core tube after sampling was measured and included in the volume calculation of the core sediment.

$$BD_2[10^3\ kg\ m^{-3}] = \frac{dry\ weight}{volume} \tag{3}$$

where dry weight is given in [g] and volume in $[cm^3]$, respectively in $[10^3\ kg]$ and $[m^3]$.

## 2.5 Radiocarbon dating

Macrofossils of 15 samples were dated for accelerator mass spectrometry (AMS) radiocarbon dating in the Radiocarbon Laboratory in Poznan, Poland (Goslar et al., 2004). The radiocarbon ages were calibrated using the CALIB 7.1 software and the IntCal13 calibration curve (Stuiver et al., 2017). All calibrated ages are expressed as calibrated kilo years before present

(cal ka BP).





### 2.6 Nitrogen and carbon content and composition

Homogenized samples were analyzed for total nitrogen (TN), total carbon (TC) and total organic carbon (TOC) using an elemental analyzer (Elementar Vario EL III and Elementar Vario Max C) and are expressed in [wt%]. The TOC/TN (weight) ratio was calculated and will be referred to as C/N ratio. The stable carbon isotopic composition was determined by

measuring $\delta^{13}C$ (ThermoFisher Scientific Delta-V-Advantage gas mass spectrometer equipped with a FLASH elemental analyzer EA 2000 and a CONFLO IV gas mixing system). The ratio is compared to the standardized Vienna Pee Dee Belemnite (VPDB) and expressed in per mille (‰ vs. VPDB). For samples with a TOC below the analytical accuracy (0.1 wt%), no C/N ratio nor $\delta^{13}C$ was measured.

    The non-parametric Mann-Whitney-Wilcoxon test was performed to test for significant differences of the

biogeochemical parameters between the stratigraphic landscape units. When the p-value exceeds 0.05, the null hypothesis ($H_0$: There is a statistically significant difference between the samples of the different stratigraphic landscape units) cannot be rejected. More extensive method descriptions of the elemental analysis can be found in Supplement 1.2.

### 2.7 Lipid biomarker analysis

#### 2.7.1 Extraction and separation

Biomarker analysis was carried out to identify the molecular composition of the OC in order to analyze the OC composition and quality (Andersson and Meyers, 2012; Strauss et al., 2015). In total, 13 samples (6 from BAL16-B2 and 7 from BAL16-B4) were analyzed at the German Research Centre for Geosciences (GFZ) for *n*-alkanes (long-chained, single bonded hydrocarbons) and branched glycerol diakyl glycerol tetraethers (brGDGTs; bacterial membrane lipids). The methods were adapted from Schulte et al. (2000) and Strauss et al. (2015). About 8 g of each sample was extracted

(Dionex 200 ASE extractor) using dichloromethane/methanol (DCM/MeOH) (99:1 v/v) (heating phase 5 min, static phase 20 min at 75°C and 106 Pa). Excess solvent was evaporated under $N_2$. A known amount of four internal standards was added: 5α-androstane, ethylpyrene, 5α-androstan-17-on and erucic acid. The samples were passed over a sodium sulfate column with *n*-hexane prior to separation by medium-pressure liquid chromatography (MPLC) (Radke et al., 1980) into three fractions: aliphatic hydrocarbons, aromatic hydrocarbons and nitrogen-, sulfur-, and oxygen (NSO) containing

compounds. The dissolved extracts were injected into the MPLC system where they were led through a column and pre-columns (thermally deactivated silica 100 63-200 μm and 200-500 μm on top with ratio ~7:1) with *n*-hexane. The NSO-fraction was further separated manually into a polar and acid fraction using a KOH-impregnated silica gel column with DCM.

#### 2.7.2 Measurements

The *n*-alkanes were measured as part of the aliphatic fraction using gas chromatography (GC) - mass spectrometry (MS) (Trace GC Ultra and MS DSQ, Thermo Electron Corporation) using helium as a carrier gas. The samples were vaporized



(50°C to 300°C with 10°C/s, 10 min isothermal holding) and led through a capillary column (BPX5; 22 mm x 50 m, film thickness 0.25 μm) (Peters et al., 2005). The oven was programmed from 50°C to 310°C (3°C/min and 30 min isothermal holding). The GC was linked to the MS to enable compound identification (ionization mode at 70 eV, 230°C). Full scan mass spectra were obtained from m/z 50 to 600 Da (2.5 scans/s). Using the software XCalibur, the peaks in the

GC-MS total ion current chromatogram were integrated manually. The *n*-alkanes were quantified by comparing the peak area of the target compounds to the applied internal standards.

The brGDGTs were measured as part of the acid fraction using high-performance liquid chromatography (Shimadzu LC-10AD HPCL device coupled to a Finnigan TS 7000 mass spectrometer with APCI interface). The compounds were separated at 30°C over a Prevail Cyano column (2.1x150 mm, 3μm; Alltech) preceded by a precolumn filter of the same material,

which does not separate 5- and 6-methyl isomers. Each fraction was eluted isocratically with *n*-hexane A and isopropanol B (5 min. 99% A and 1% B, linear gradient to 1.8% B within 40 min, in 1 min to 10% B holding for 5 min and back to initial conditions in 1 min, held for 16 min) with a flow rate of 0.2 mL/min. The APCI device has a corona current of 5 μA, voltage of 5 kV, the vaporizer temperature is 350°C and the capillary temperature 200°C. The source operates with nitrogen sheath gas at 60 psi without auxiliary gas. Full mass spectra were obtained at a scan rate of 0.33 s. The integration was performed in

XCalibur and the quantification was performed by comparing the compound peaks to an Archaeol run.

### 2.7.3 Biomarker derived indices

The carbon preference index (CPI) of the *n*-alkanes was calculated after Marzi et al. (1993) following Eq. (4), and the average chain length (ACL) after Poynter and Eglinton (1990) following Eq. (5).

$$CPI_{23-33} = \frac{\sum_{i=n}^{m} C_{2i+1} + \sum_{i=n+1}^{m+1} C_{2i+1}}{2*(\sum_{i=n+1}^{m+1} C_{2i})} \tag{4}$$

where n is the starting dominating chain length divided by 2, m the ending dominating chain length divided by 2 and i the carbon number index.

$$ACL_{23-33} = \frac{\sum i C_i}{\sum C_i} \tag{5}$$

where i is the carbon number and C the concentration. For the calculation of both CPI and ACL, the interval of $C_{23}$ to $C_{33}$ was used. The ACL indicates OC source, where higher land plants (i.e. vascular plants) are dominated by long-chain

*n*-alkanes of 25 to 30 carbon atoms, whereas bacteria and algae contain mainly shorter chains of 15 to 20 carbon atoms (Killops and Killops, 2013; Strauss et al., 2015). The CPI is the odd-to-even predominance of the hydrocarbons and indicates the degree of degradation of the OC, where a lower value indicates further degraded material (Andersson and Meyers, 2012; Glombitza et al., 2009).

### 2.8 Landscape organic carbon pool estimation

A map of key landscape units (yedoma hills, DTLBs and thermokarst lakes) of the northern part of Baldwin Peninsula was developed in order to calculate the coverage for each landscape unit and to allow a first-order estimate of the belowground





OC storage. Using a Landsat 8 satellite image (false color image with short-wave infrared, near-infrared and deep blue bands (bands 7-5-1), pixel resolution 30 m) as well as a digital terrain model (DTM; grid cell resolution 5 m), the three main landscape units were manually mapped and digitized for the northern part of the Baldwin Peninsula (~450 km$^2$). A similar approach of operator-driven thermokarst mapping by manually digitizing landforms from remote sensing imagery in

combination with a DTM was successfully applied by Morgenstern et al. (2011) and Farquharson et al. (2016).

The total OC pool on Baldwin Peninsula was then estimated based on the deposit thickness, the spatial coverage, wedge ice volume (WIV), BD and TOC from the sampled exposures and cores (BAL16-B2 to B5 and BAL16-UPL1-L1) following Eq. (6).

$$Total\ OC\ pool\ [Mt] = \frac{thickness\ *coverage\ *\frac{100-WIV}{WIV}*BD*\frac{TOC}{100}}{10^6} \qquad (6)$$

where the deposit thickness is given in [m], spatial coverage in [m$^2$], WIV in [vol%], BD in [$10^3$ kg m$^{-3}$] and TOC in [wt%]. The deposit thickness is based on few field observations and for this first-order assessment assumed constant over the whole peninsula. Deeper deposits below frozen yedoma and DTLB are excluded due to unknown spatial coverage and thickness. Also, the deposits below lake sediments, potentially unfrozen in a talik, are not included in the calculation. The spatial coverage per landscape unit was calculated from the land coverage classification map. Strauss et al. (2013) calculated

average wedge-ice volume for yedoma and DTLB deposits based on polygon size and ice-wedge width and depth. WIV was assumed to be zero for the thermokarst lake sediments. In order to compensate for possibly non-continuous field sampling, weighted BD*TOC values over depth were used by value replication depending on depth interval. The data were extrapolated to the northern part of the Baldwin Peninsula (i.e. the mapped part). The volumetric OC pool was estimated according to Eq. (7).

$$Volumetric\ OC\ pool\ [kg\ m^{-3}] = \left(\frac{100-WIV}{WIV}*BD*\frac{TOC}{100}\right)*10^3 \qquad (7)$$

The calculation for each landscape unit was performed separately using bootstrapping techniques, which included $10^4$ iterations of random sampling with replacement, after which the mean and standard deviation were calculated. Because BD and TOC are correlated, paired values were used during the random sampling.

### 3 Results

### 3.1 Biogeochemical and biomarker proxies

### 3.1.1 Yedoma exposure

Radiocarbon dates for yedoma exposure BAL16-B2 range from > 50,000 to 10,000 cal a BP (Table 1). There is no consistent age-depth relationship: while a few samples, including the near-surface sample, have an infinite age, the second youngest sample is found at a depth of 1600 cm. The cryostratigraphical and biogeochemical parameters are presented by depth

(Figure 2). Depths are measured from the cliff top downwards for the yedoma and DTLB exposures, and from the sediment



surface downwards for the thermokarst lake sediments. The BD of the yedoma deposits (mean: $0.80*10^3$ kg m$^{-3}$, sd: 0.18) shows most variation in the bottom part up to 1117 cm. The TOC ranges between 0.1 and 4.7 wt% (mean: 1.9 wt%, sd: 1.1). The C/N ratio varies between 4.4 and 14.0 (mean: 10.1, sd: 2.9) and does not show a trend with depth. The $\delta^{13}$C is in the range of -24.8 to -28.3 ‰ (mean: -25.9 ‰, sd: 0.9) and shows most variation up to 1080 cm. Exposure BAL16-B2 is very ice-rich with an average absolute ice content of 45 wt% (70 vol%).

The biomarkers concentrations and index values are presented by depth in Figure 3. The $n$-alkane concentration in the yedoma deposits ranges from 0.5 to 1.8 μg g$^{-1}$ TOC (mean: 1.0 μg g$^{-1}$ TOC, sd: 0.5). The $ACL_{23-33}$ varies between 28.8 in the bottom sample and 28.2 in the top with a peak of 30.0 at 1117 cm (mean: 28.8, sd: 0.6). The samples are dominated by $n$-alkane chains with a high number of carbon atoms: the dominating chains are $n$-$C_{29}$ and $n$-$C_{31}$ (Figure 4). The $CPI_{23-33}$ is in the range of 9.0 to 13.6 (mean: 11.6, sd: 2.0) and does not show a trend over depth. The sample at 1399 cm has the maximum brGDGT concentration of 1.3 mg g$^{-1}$ TOC whereas the other samples have relatively low concentrations (mean: 0.5 mg g$^{-1}$ TOC, sd: 0.5).

### 3.1.2 DTLB exposure

Radiocarbon ages for DTLB exposure BAL16-B4 are in the range from > 50,000 cal a BP (788 to 434 cm) to 240 cal a BP (22 cm) and show no age inversions: the lower samples are of infinite age and the near-surface sediments are the youngest and show Holocene ages. The BD decreases upwards in the profile and ranges between 1.27 and $0.29*10^3$ kg m$^{-3}$ (mean: $0.91*10^3$ kg m$^{-3}$, sd: 0.3), a particularly strong decrease occurs from 280 cm to 215 cm. The TOC (mean: 6.6 wt%, sd: 9.4) is higher in the interval from 250 to 22 cm than in the lower part, and strongly increases in the top 8 cm. The C/N ratio increases towards the top, ranging from 11.3 at the bottom to 29.0 at the top (mean: 14.9, sd: 4.2). The $\delta^{13}$C displays an opposite trend with higher values in the lower part (-25.8‰) and a decrease near the surface (-28.7‰) (mean: -27.2‰, sd: 0.7), and a sudden decrease at 280 to 250 cm. The exposure has an average absolute ice content of 41 wt% (67 vol%).

The $n$-alkane concentration reaches its maximum of 16.0 μg g$^{-1}$ TOC in the bottom sample whereas the other samples have much lower concentrations (mean: 4.3 μg g$^{-1}$ TOC, sd: 5.4) (Figure 3). The $ACL_{23-33}$ is higher than 28 for all samples except at 583 cm and 430 cm (mean: 28.3, sd: 0.4). The dominating $n$-alkane chains are $n$-$C_{27}$ and $n$-$C_{31}$ (Figure 4). The $CPI_{23-33}$ increases towards the top from 5.7 to 12.6 (mean: 8.8, sd: 2.1). The brGDGT concentration varies between 0 and 3.8 mg g$^{-1}$ TOC with the peak at 160 cm below the surface (mean: 1.5 mg g$^{-1}$ TOC, sd: 1.4).

### 3.1.3 Thermokarst lake sediments

The radiocarbon age of the thermokarst lake core BAL16-UPL1-L1 is 2,010 cal a BP at a depth of 26.5-27.5 cm and 480 cal a BP at 19-20 cm. The sedimentological and biogeochemical parameters have a low variability along the core profile with a mean for BD of $0.78*10^3$ kg m$^{-3}$ (sd: 0.01), TOC of 14.4 wt% (sd: 0.5), C/N ratio of 22.5 (sd: 0.6) and $\delta^{13}$C of -28.5 ‰ (sd: 0.2). The $\delta^{13}$C shows a slight decrease towards the lake sediment surface.





### 3.1.4 Statistical significance

We found significant differences ($p<0.05$) for all pairwise comparisons of the biogeochemical parameters between the yedoma, DTLB and thermokarst lake deposits on Baldwin Peninsula (Supplement 2.4). The DTLB deposits have the highest BD and the thermokarst lake sediments the lowest. The TOC values and C/N ratios are highest in the thermokarst lake sediments followed by the DTLB and then the yedoma deposits, whereas the $\delta^{13}C$ values show an opposite trend.

### 3.2 Organic carbon pool estimation

We produced a land cover classification map distinguishing between yedoma, DTLBs, thermokarst lakes and lagoons (Figure 5). For the lakes on yedoma uplands and lagoons, no field information is available, so that those areas are excluded from further OC pool calculations. The total area mapped is about 450 km$^2$ of which ~65% is covered by DTLB, ~30% by yedoma and ~5% by thermokarst lakes (Table 2). The input parameters of the bootstrapping for the OC pool estimates are shown in Table 2. The volumetric and total OC pool per stratigraphic landscape unit for the frozen deposits and unfrozen thermokarst lake sediments are presented in Table 2, where the WIV is included. The OC pool estimates without WIV are also reported to allow comparison with other studies (Table 2). The yedoma deposits contain $8.0\pm0.4$ kg OC m$^{-3}$ ($15.3\pm0.8$ excl. WIV) and the DTLB deposits $34.7\pm1.4$ kg OC m$^{-3}$ ($37.4\pm1.6$ excl. WIV), which corresponds to a total OC pool of $16.3\pm0.9$ Mt ($31.4\pm1.7$ excl. WIV) in yedoma and $51.5\pm2.3$ Mt ($55.4\pm2.3$ excl. WIV) in DTLB deposits. The total estimate of the total OC pool of the frozen sediments on northern Baldwin Peninsula is ~68 Mt OC. The thermokarst lake sediments contain $92.9\pm0.4$ kg OC m$^{-3}$ which adds up to a total pool of $3.9\pm0.0$ Mt for all thermokarst lakes in the study area.

## 4 Discussion

### 4.1 Landscape development and carbon dynamics

#### 4.1.1 Sediment facies

The yedoma deposits of BAL16-B2 have been accumulated in a stable predominantly aeolian depositional environment, as shown by the grain size distribution (detailed method description and results of the grain size analysis are given in Supplement 1.1 and 2.3.1). The grain size distributions indicate that these deposits are characterized by a stronger aeolian influence than northeastern Siberian yedoma sites (Schirrmeister et al., 2008b; Strauss et al., 2012). Field observations suggest that the lower part of exposure BAL16-B2 (1,600 to 1,870 cm) is a separate unit. The significant change in sediment type from very coarse silt to medium silt (p-value<0.05), reflected in the grain size distributions (Supplement 2.3.1), confirms this distinction.

The formation of yedoma deposits on the west coast of the Baldwin Peninsula at study site BAL16-B2 possibly started before 50 cal ka BP. The large range of radiocarbon ages in these yedoma deposits suggests that they are mixed with ancient or younger organic material. According to Vasil'chuk and Vasil'chuk (2017), contamination with ancient organic material is





common in yedoma deposits, due to the syngenetic character of the deposits. Therefore, they proposed to take the youngest age of the sediments, justified by the fact that age rejuvenation is not likely, due to the undisturbed character of the deposits. However, it is unlikely that relatively young sediments (~17 cal ka BP at 1600 cm) are overlain by a few meters of older sediments (~45 cal ka BP at 1117 cm and infinite ages at 1399 and 1500 cm), suggesting that the young ages could be the

result of re-deposition. Considering that BAL16-B2 is a coastal bluff, re-deposition of young material cannot be neglected. Furthermore, the ages fall in the range of previous studies from Siberian yedoma deposits (> 57 to 13 ka BP) (Schirrmeister et al., 2002a, 2002b, 2003; Strauss et al., 2013) and Alaskan yedoma deposits (> 48 to 14 ka BP) (Kanevskiy et al., 2011), except for the sample at 945 cm (~10 cal ka BP). Generally, care should be taken with the interpretation of dating of yedoma deposits.

The lower part of the DTLB exposure BAL16-B4 has a radiocarbon age of 46 cal ka BP or older, which suggests that the lowest deposits have been accumulated during the same time as yedoma formation. However, the TOC content and C/N ratio are significantly lower in BAL16-B2 than in BAL16-B4 whereas the $\delta^{13}$C is significantly higher (Supplement 2.4). This suggests that the DTLB deposits are a mixture of former yedoma deposits thawed and partially decomposed in a lake talik, lake sediments and post-drainage terrestrial peat. The grain size signal for both exposures is very similar (p>0.05),

suggesting a connection between the BAL16-B2 and BAL16-B4 sediments, which can be explained by the Holocene reworking of the yedoma sediments in the lake and in the talik affected by subsidence. The branched and isoprenoid tetraethers (BIT) index (detailed method description and results of the biomarker climatic indicator are given in Supplement 1.3 and 2.3.2) is lower for BAL16-B2 than for BAL16-B4, which may indicate that the sediments in BAL16-B2 have been deposited in a drier climate (Dirghangi et al., 2013), in line with earlier paleoenvironmental reconstructions (e.g. Andreev et

al., 2011; Lenz et al., 2016c).

In exposure BAL16-B4 a change was observed in the proxy records (age, BD, TOC, $\delta^{13}$C) between 280 and 250 cm: the sediments above 250 cm have lower BD and $\delta^{13}$C and a higher TOC and C/N ratio compared to the sediments below 280 cm. Furthermore, there is a change in sediment type, as shown by the different grain size distributions below 280 and above 250 cm (Supplement 2.3.1), and in the depositional environment, as shown by a change in the relative brGDGT distribution,

reflected in the methylation of branched tetraethers (MBT) index (Supplement 1.3 and 2.3.2). These changes suggest the initiation of a thermokarst lake: the sediments below 280 cm likely are former yedoma sediments that were thawed in situ in a talik (taberite), whereas the upper sediments are lake sediments. The lake initiation likely happened during the Holocene. The drainage of the lake is not recorded in the data; the drainage event took place after 2.08 cal ka BP.

The thermokarst lake sediments from BAL16-UPL1-L1 have been accumulated during the Holocene (2,010 and 480 a BP).

Assuming a recent age for the surface sediments, the calculated sedimentation rate between 2,010 and 480 years BP was ~6 cm ka$^{-1}$ and the sedimentation rate for the last 480 years ~50 cm ka$^{-1}$. For comparison, Lenz et al. (2016b) found a mean sedimentation rate for Peatball Lake, Alaska, of ~70 cm ka$^{-1}$ for the last 1,400 years. The BD, TOC, C/N ratio and $\delta^{13}$C of the sediment core BAL16-UPL1-L1 show minimal variation, suggesting a stable depositional environment. Therefore, a hiatus in the sediment is unlikely.



### 4.1.2 Organic carbon quantity

The TOC content of the yedoma deposits is comparable to TOC values reported in previous yedoma studies from across Siberia and Alaska (Figure 6). The thermokarst lake sediment has the highest TOC of the three landscape units, which is likely due to the addition of organic matter from lake primary production, as well as the integration of organic matter from
its catchment. The DTLB deposits cover a large range of TOC values, which corroborates the findings from section 4.1.1 that the deposits are a mixture of reworked and new material. Although the mean TOC content of the yedoma deposits is relatively low compared to that of DTLB deposits and thermokarst lake sediments, the spatial coverage and thickness of the yedoma deposits is large and thus a significant OC pool is still expected.

Thermokarst lake sediments have the highest volumetric OC pool compared to yedoma deposits and DTLB deposits. The
DLTB deposits ($\sim$35 kg m$^{-3}$) contain four times as much OC by volume as yedoma deposits ($\sim$8 kg m$^{-3}$). However, the volumetric OC pool of the DTLB deposits ($\sim$37 kg m$^{-3}$) is more than twice as high as the yedoma OC pool ($\sim$15 kg m$^{-3}$), when excluding WIV. Multiple studies have been carried out to estimate OC pool size of deep yedoma and DTLB deposits (Schirrmeister et al., 2011; Shmelev et al., 2017; Strauss et al., 2013; Zimov et al., 2006). Schirrmeister et al (2011) (Northeast Siberia) and Strauss et al. (2013) (total yedoma region, $\sim$1,387,000 km$^2$) both found that DTLB deposits contain
approximately three times as much OC as yedoma deposits (Figure 7) (including WIV). Zimov et al. (2006) found that yedoma deposits contain on average 18.5 kg m$^{-3}$ (excluding WIV). On the other hand, Webb et al. (2017) also estimated the OC pool of Siberian deep deposits (0-15 m) and found that the yedoma deposits contained more OC (7.9 to 21.6 kg m$^{-3}$) than DTLB deposits (6.9 to 14.5 kg m$^{-3}$) (excluding WIV). Other studies in Siberia (Fuchs et al., 2018; Siewert et al., 2016) also found more OC stored in yedoma deposits than in DTLB deposits. However, the estimates of these studies were based on
near-surface sediments (0-2 m).

Based on our landscape unit map we scaled the total OC pool of the Baldwin Peninsula. About 70% of the area of Baldwin Peninsula is affected by permafrost degradation (e.g. thermokarst processes or coastal erosion) and is therefore classified as thermokarst. Moreover, these thermokarst processes led to more than 10 m of ground subsidence on the Baldwin Peninsula, as suggested by relief differences between yedoma uplands and DTLB. The estimated OC pool of the
frozen sediments (yedoma 0-15 m depth, DTLB 0-5 m depth) on Baldwin Peninsula is $\sim$68 Mt, of which roughly three quarters is stored in DTLB deposits and one quarter in yedoma deposits. This high amount of OC stored in an area of approximately 450 km$^2$ shows the important contribution of these deep thermokarst affected yedoma deposits to pan-arctic soil organic carbon stock estimations.

### 4.2 Organic carbon quality

The source of OC influences both the quantity as well as the quality of OC. The ACL for both the yedoma ($>$28.2) and DTLB ($>$27.7) deposits suggests that the organic matter is mainly derived from terrestrial higher plants. Additionally, in all



samples, the long- and odd-chains dominate, which is also typical for a terrestrial higher plant origin (Eglinton and Hamilton, 1967).

The combination of the C/N ratio and $\delta^{13}$C value has been widely used as indicator of OC source. A higher C/N ratio suggests an enhanced input of terrestrial land plants whereas algal produced matter is generally characterized by a lower C/N

ratio. The C/N ratio and $\delta^{13}$C value of OC furthermore allow to distinguish between marine and lacustrine algae, where marine algae generally have higher $\delta^{13}$C values (e.g. Lenz et al., 2016a; Meyers, 1994, 1997). Moreover, the C/N ratio and $\delta^{13}$C value have been used as an indicator of OC decomposition, where a lower C/N ratio and higher $\delta^{13}$C value indicate further degraded material (i.e. lower quality) (e.g. Gundelwein et al., 2007; Schädel et al., 2014).

Figure 8 shows a scatter plot of the C/N ratio and $\delta^{13}$C for the three landscape units of this study (BP) and other studies

from Alaska: yedoma deposits along the Itkillik River (IR) (Lapointe et al., 2017; Strauss et al., 2012), DTLB deposits from the Northern Seward Peninsula (NSP) (Lenz et al., 2016c) and thermokarst lake sediments from lakes in the Kobuk River Delta (KOB) and Central Seward Peninsula (CSP) (Lenz et al., 2018). We found significant statistical differences (p<0.05) in almost all tests attempting to differentiate between OC composition in yedoma, DTLB and thermokarst lake deposits of the Baldwin Peninsula and the other Alaskan studies based on the C/N ratio and the $\delta^{13}$C value (Supplement 2.4). A trend

exists from low C/N ratios and high $\delta^{13}$C values in yedoma deposits towards high C/N ratios and low $\delta^{13}$C values in the thermokarst lake sediments (Figure 8). Intermediate C/N ratios and $\delta^{13}$C values were found for DTLB deposits. The high $\delta^{13}$C signal and low C/N ratio for the yedoma exposure (BAL16-B2) is typical (Dutta et al., 2006; Sánchez-García et al., 2014; Vonk et al., 2013), since it is characteristic for stadial periods with decreased productivity and a dry climate (Schirrmeister et al., 2011, 2013). The C/N ratio and $\delta^{13}$C value of the DTLB deposits show a large range, which

reflects the mixed character of the disturbed landscape: these deposits contain on the one hand reworked OC (e.g. from former yedoma deposits) and on the other hand fresh OC (e.g. from thermokarst lake production). OC in thermokarst lake sediments is generally characterized by a low $\delta^{13}$C value (Cohen, 2003; Meyers, 1994), which is also the case here (Figure 8). The high C/N ratio of the thermokarst lake sediments likely is an indication that the OC is of high quality due to contribution of fresh OC (Schirrmeister et al., 2013; Strauss et al., 2015). Hence, the trends of the C/N ratio and $\delta^{13}$C values

(Figure 8) can be explained by a combination of the source and quality of OC.

Comparing the properties of the OC pools per landscape unit at Baldwin Peninsula with the Alaskan study sites in Figure 8 shows that the yedoma from Baldwin Peninsula does not have a significantly different C/N ratio or $\delta^{13}$C value than that at Itkillik River (p>0.05; Supplement 2.4) which suggests that the OC in both yedoma deposits has a similar composition. The DTLB deposits of Baldwin Peninsula have a significantly higher C/N ratio (p<0.05) compared to Northern Seward

Peninsula, but a similar $\delta^{13}$C (p>0.05). The lower C/N ratio at Northern Seward Peninsula can be explained by the multiple lake generations this basin went through, leading to generally more degraded OC (Lenz et al., 2016a). Given that the C/N ratios and $\delta^{13}$C values among thermokarst lakes at Baldwin Peninsula and Central Seward Peninsula are significantly indistinguishable (p>0.05), we argue that general contributions of lake production and thawed sediments from below are comparable, resulting in a similar OC composition. The thermokarst lake in Kobuk River Delta, however, is significantly



different in both proxies compared to Baldwin Peninsula and Central Seward Peninsula (p<0.05). The lower $\delta^{13}$C value the Kobuk River Delta can be an indication that the lacustrine contribution to the OC pool is higher than in Baldwin Peninsula and Central Seward Peninsula. Nonetheless, the thermokarst lake sediments from Baldwin Peninsula, Kobuk River Delta and Central Seward Peninsula all have relatively high C/N ratios and $\delta^{13}$C values, compared to the yedoma and DTLB deposits (Figure 8).

A significantly higher CPI in yedoma compared to DTLB deposits (p-value<0.05) indicates that the OC in the yedoma deposits is less degraded and, therefore, suggests that this pool has a higher potential for future OC decomposition. Furthermore, the decreasing trend in CPI in the DTLB deposits indicates progressive degradation with depth, whereas the lack of such a trend in the yedoma deposits indicates that minimal or no decomposition has occurred before the organic matter was incorporated into the permafrost, maintaining its high quality (Stapel et al., 2016; Strauss et al., 2015; Weiss et al., 2016). These findings suggest that the OC in yedoma deposits BAL16-B2 is of higher quality than that in the DTLB deposits BAL16-B4.

Regardless, to evaluate implications of permafrost degradation arising from future climate change, it is necessary to assess the vulnerability of the total OC pool. Climate change will increase the frequency and intensity of fires and floods which can lead to soil removal and disturbances of the ground thermal regime which can result in rapid local permafrost degradation (Grosse et al., 2011). Because of the high ice content in the yedoma and DTLB deposits on Baldwin Peninsula, the deposits are highly susceptible and vulnerable to deep permafrost thaw which will have a great effect on the topography. With the formation of new, deep lakes, primary productivity is expected to increase on a large scale, which will possibly compensate for increased greenhouse emissions initially. Lindgren et al. (2016) found a net increase in total lake surface area by 3.9% for the Baldwin Peninsula between 1972 and 2014. However, following the positive permafrost feedback loop, more intensive permafrost degradation will likely lead to an increase of the number of lake drainage events in West-Alaska (Lindgren et al., 2016). This could lead to rapid OC sequestration in new permafrost aggrading in DTLBs and evolving terrestrial peat, but will be followed by ultimate decomposition once this region is affected by widespread permafrost near-surface thaw between 2050 and 2100 (Lawrence and Slater, 2005; Walter Anthony et al., 2014). Although the largest share of OC in frozen deposits on Baldwin Peninsula is stored in DTLB deposits (~75%), the OC in yedoma deposits is of higher quality and therefore, the OC in the yedoma is especially vulnerable to future microbial degradation and greenhouse gas release which will further enhance climate warming.

## 5 Conclusion

This study presents OC characteristics from ice-rich permafrost deposits and thermokarst lake sediments on the Baldwin Peninsula, West-Alaska. Using cryostratigraphical, biogeochemical and biomarker parameters of yedoma deposits and DTLB deposits as well as thermokarst lake sediments, the size and quality of OC pools in ice-rich permafrost were identified. The OC pool of the frozen deposits in a 450 km² study on the Baldwin Peninsula is ~68 Mt, of which three





quarters are stored in frozen DTLB deposits. The lake sediments had the highest volumetric OC content of ~93 kg OC m$^{-3}$ compared to yedoma (~8 kg OC m$^{-3}$) and DTLB deposits (~35 kg OC m$^{-3}$). Biogeochemical and biomarker parameters indicated that the OC in the yedoma deposits is best preserved and of higher quality than the OC stored in DTLB deposits and thermokarst lake sediments, demonstrating the higher potential for OC decomposition in yedoma deposits.

## 5 Data publishing

The data presented in this study will be available on PANGAEA after submission (Jongejans et al., 2018).

## Acknowledgements

This study was carried out within the ERC PETA-CARB project (#338335) and additional support by the Helmholtz Impulse and Networking Fund (#ERC-0013). L.L. Jongejans was financially supported through an Erasmus+ EU grant. Field work
was carried out by the Alfred Wegener Institute, Helmholtz Centre for Polar and Marine Research and the U.S. Geological Survey (USGS). We thank the Kotzebue Community, Ben Jones (USGS), Jim Kincaid (Northwestern Aviation), Jim Webster (Webster's Flying Service) and Ingmar Nitze (AWI) for help in the field, and Dyke Scheidemann (AWI), Anke Sobotta and Cornelia Karger (German Research Centre for Geosciences) for their analytical support in the lab.

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

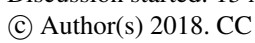



**Table 1: Radiocarbon dates of BAL16-B2, BAL16-B4 and BAL16-UPL1-L1. Calibrations were performed using CALIB 7.1 software and the IntCal13 calibration curve (Stuiver et al., 2017). Poz: Poznan Radiocarbon Laboratory, Poland, pMC: percent modern carbon.**

| Sample ID | Ext. ID | Depth [cm] | $^{14}$C ages [a BP] | ± | Mean cal ages 2σ (95.4 %) [a BP] | ± | Rounded $^{14}$C ages [cal a BP] | ± |
|---|---|---|---|---|---|---|---|---|
| BAL16-B2-20 | Poz-89527 | 620 | > 48,000 | | | | | |
| BAL16-B2-26 | Poz-89700 | 945 | 8,890 | 50 | 10,037.5 | 153.5 | 10,000 | 150 |
| BAL16-B2-31 | Poz-89702 | 1117 | 40,810 | 1800 | 44,747 | 3125 | 44,700 | 3,130 |
| BAL16-B2-39 | Poz-89703 | 1399 | > 50,000 | | | | | |
| BAL16-B2-1 | Poz-89526 | 1500 | > 50,000 | | | | | |
| BAL16-B2-5 | Poz-89523 | 1600 | 16,200 | 90 | 19,551.5 | 287.5 | 19,600 | 290 |
| BAL16-B4-2a | Poz-89704 | 22 | 105.7 | 0.33 pMC | 239 | 15 | 240 | 20 |
| BAL16-B4-4a | Poz-89705 | 132 | 1,700 | 30 | 1,590.5 | 46.5 | 1,590 | 50 |
| BAL16-B4-6a | Poz-89706 | 166 | 2,125 | 30 | 2,079.5 | 77.5 | 2,080 | 80 |
| BAL16-B4-14a | Poz-89707 | 340 | 42,800 | 1,600 | 46,361.5 | 2,967.5 | 46,400 | 3,000 |
| BAL16-B4-18a | Poz-89708 | 434 | > 50,000 | | | | | |
| BAL16-B4-24a | Poz-89709 | 583 | > 50,000 | | | | | |
| BAL16-B4-31a | Poz-89710 | 788 | > 50,000 | | | | | |
| BAL16-UPL1-L1-A | Poz-89349 | 19-20 | 425 | 30 | 482 | 43 | 480 | 40 |
| BAL16-UPL1-L1-B | Poz-89351 | 26.5-27.5 | 1,940 | 30 | 2,014.5 | 64.5 | 2,015 | 60 |




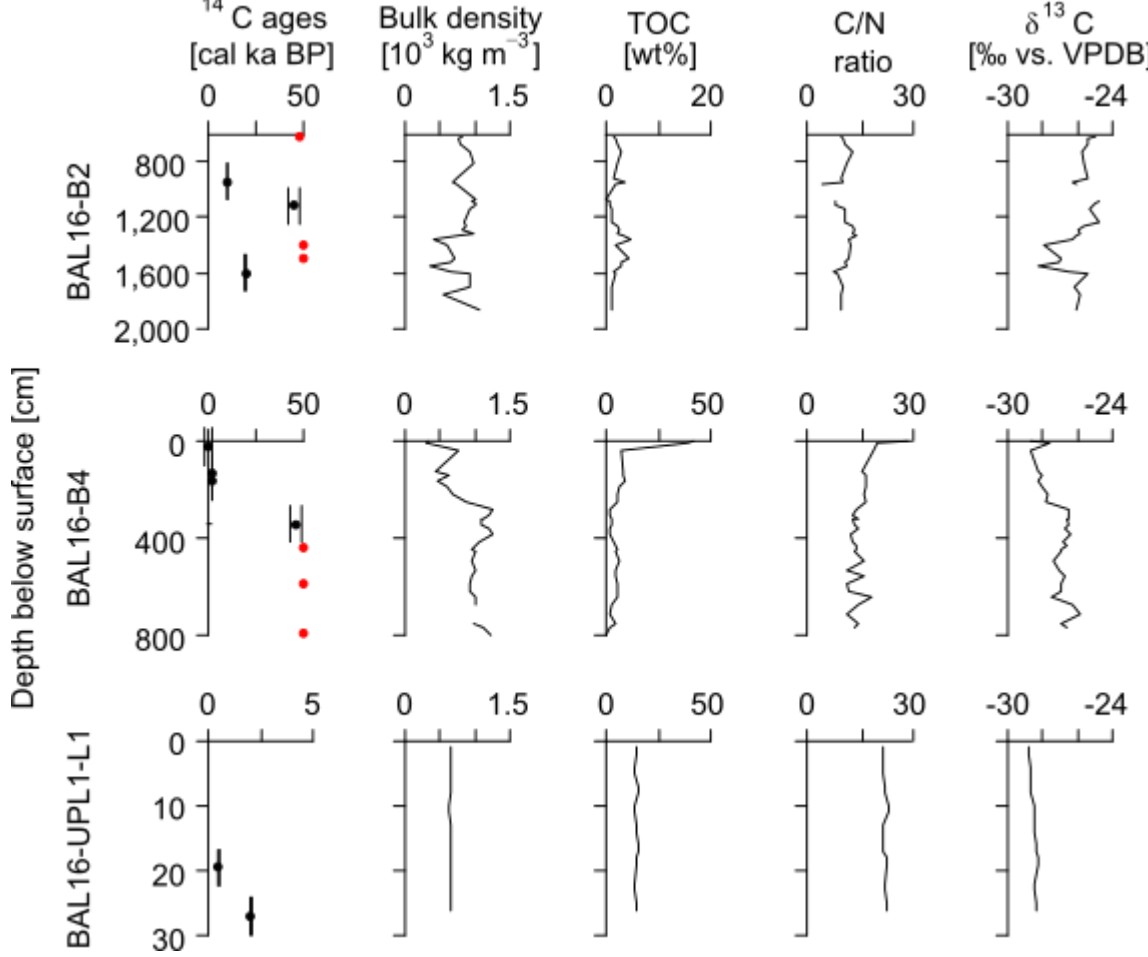

**Figure 2: Summary of cryostratigraphical and biogeochemical parameters of BAL16-B2 (yedoma), BAL16-B4 (drained thermokarst lake basin) and BAL16-UPL1-L1 (thermokarst lake): calibrated radiocarbon ages ([14]C; infinite ages in red), bulk density, total organic carbon (TOC), total organic carbon-total nitrogen (C/N) ratio, stable carbon isotopes (δ[13]C). Note: different x-axes for [14]C and TOC.**





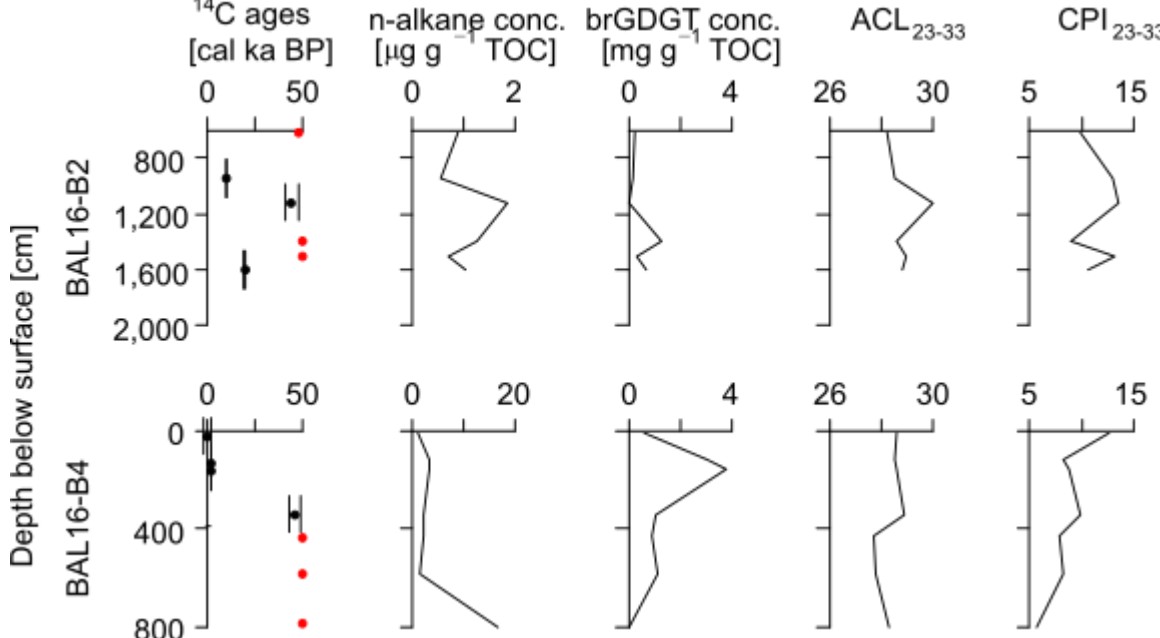

**Figure 3: Summary of biomarker parameters of BAL16-B2 (yedoma) and BAL16-B4 (drained thermokarst lake basin): calibrated radiocarbon ages ($^{14}$C; infinite ages in red), *n*-alkane concentration, brGDGT concentration, average chain length (ACL) and carbon preference index (CPI). ACL and CPI were calculated from *n*-alkane range C$_{23-33}$. Note: different x-axes for *n*-alkane concentration.**




**Figure 4:** *n*-alkane concentrations of BAL16-B2 (yedoma) and BAL16-B4 (drained thermokarst lake basin) by depth. Odd chains (grey bars) and even chains (red bars), sample depth and dominating *n*-chain indicated. Note: different y-axes.





**Figure 5: a) Land cover classification map of Baldwin Peninsula (see overview map in upper-right corner) with yedoma (green), drained thermokarst lake basins (yellow), thermokarst lakes (blue) and lagoons (grey). B) Exemplary photos: I. Yedoma (note person on the right indicated by yellow arrow for scale), II. Drained thermokarst lake basin and III. Thermokarst lake. Source for overview map in (a): World Ocean Base (ESRI).**





**Table 2: Input (left) and output parameters (right) for organic carbon pool calculations for yedoma, drained thermokarst lake basin and thermokarst lake deposits: deposit thickness, landscape coverage, wedge-ice volume (WIV), average weighted bulk density * total organic carbon (BD*TOC). WIV data from Ulrich et al. (2014).**

| Landscape unit | Thickness | Coverage | WIV | Volumetric OC pool Incl. WIV | Total OC pool Incl. WIV | Volumetric OC pool excl. WIV | Total OC pool excl. WIV |
| --- | --- | --- | --- | --- | --- | --- | --- |
| | [m] | [m$^2$] | [vol %] | [kg m$^{-3}$] | [Mt] | [kg m$^{-3}$] | [Mt] |
| Yedoma | 15 | 136,620,000 | 48 | 8.0 ± 0.4 | 16.3 ± 0.9 | 15.3 ± 0.8 | 31.4 ± 1.7 |
| DTLB | 5 | 296,440,000 | 7 | 34.7 ± 1.4 | 51.5 ± 2.2 | 37.4 ± 1.6 | 55.4 ± 2.3 |
| Thermokarst lake | 2 | 21,180,000 | 0 | | | 92.9 ± 0.4 | 3.9 ± 0.0 |

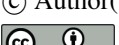


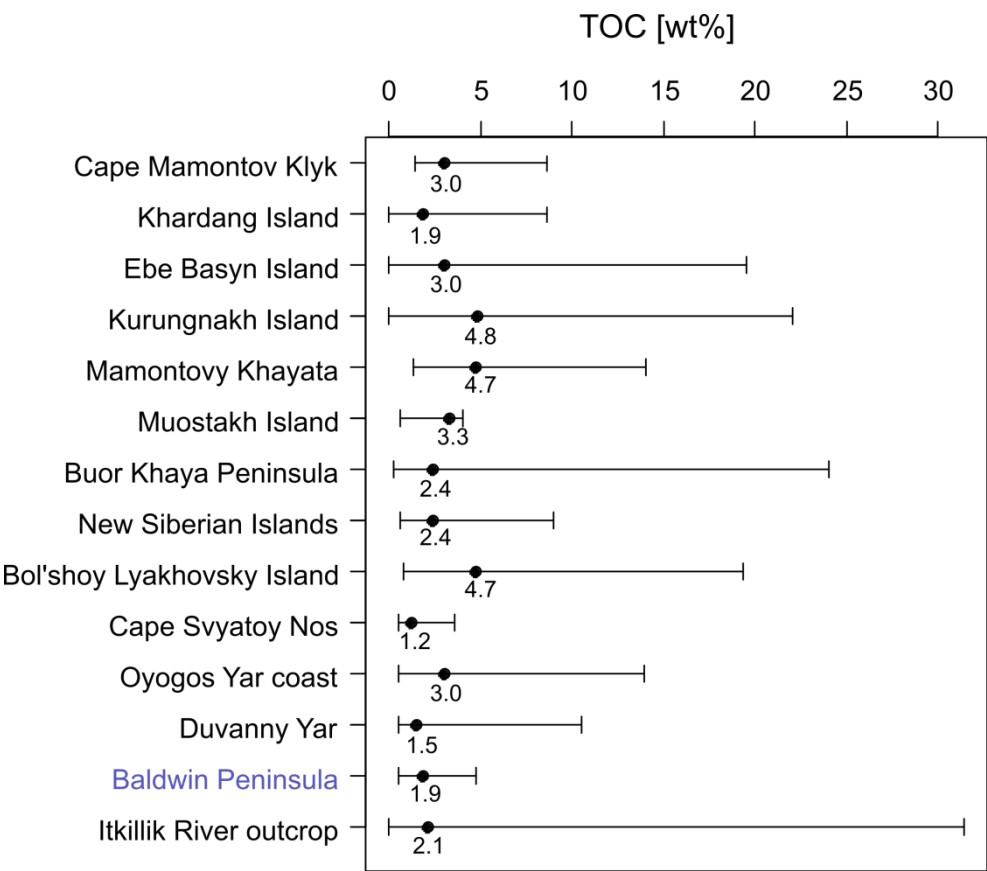

**Figure 6: TOC variations from different yedoma study sites in Siberia and Alaska. Sorted from westernmost (Cape Mamontov Klyk, western Laptev Sea) to easternmost (Itkillik River outcrop, Alaskan North Slope) study sites. Data from Schirrmeister et al. (2008a, 2008b, 2011), Strauss et al. (2012, 2013, 2015) and Baldwin Peninsula (this study; blue).**




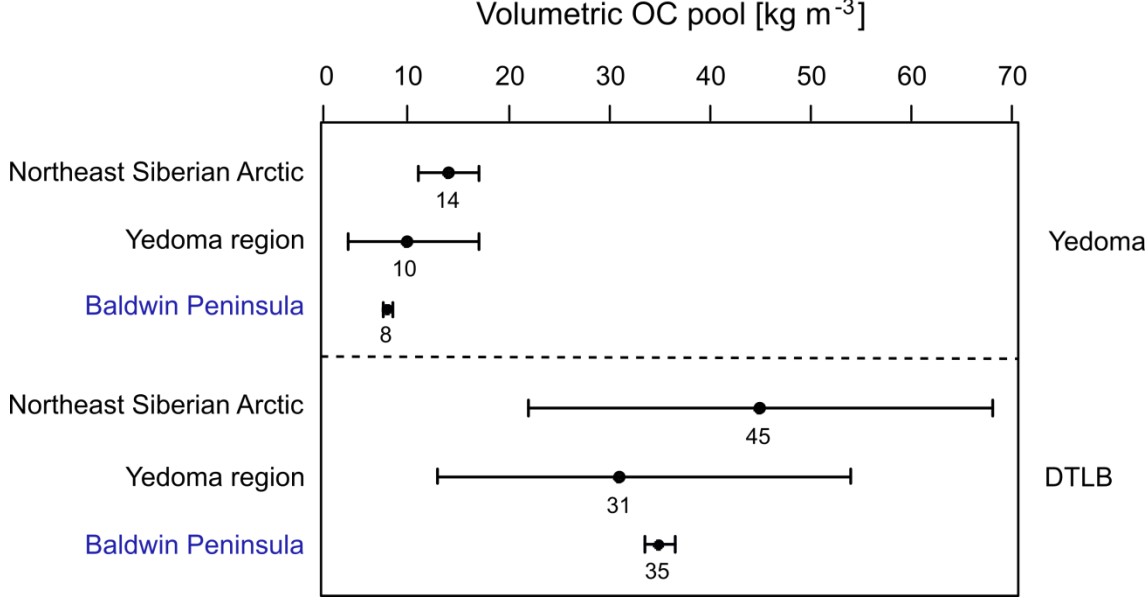

**Figure 7: Volumetric OC pool estimates of yedoma deposits (left) and drained thermokarst lake basin (DTLB) deposits (right).**
**Mean values indicated. Data from Northeast Siberian Arctic (Schirrmeister et al., 2011), the yedoma region in Siberia and Alaska**
5 **(Strauss et al., 2013) and Baldwin Peninsula (this study; blue).**



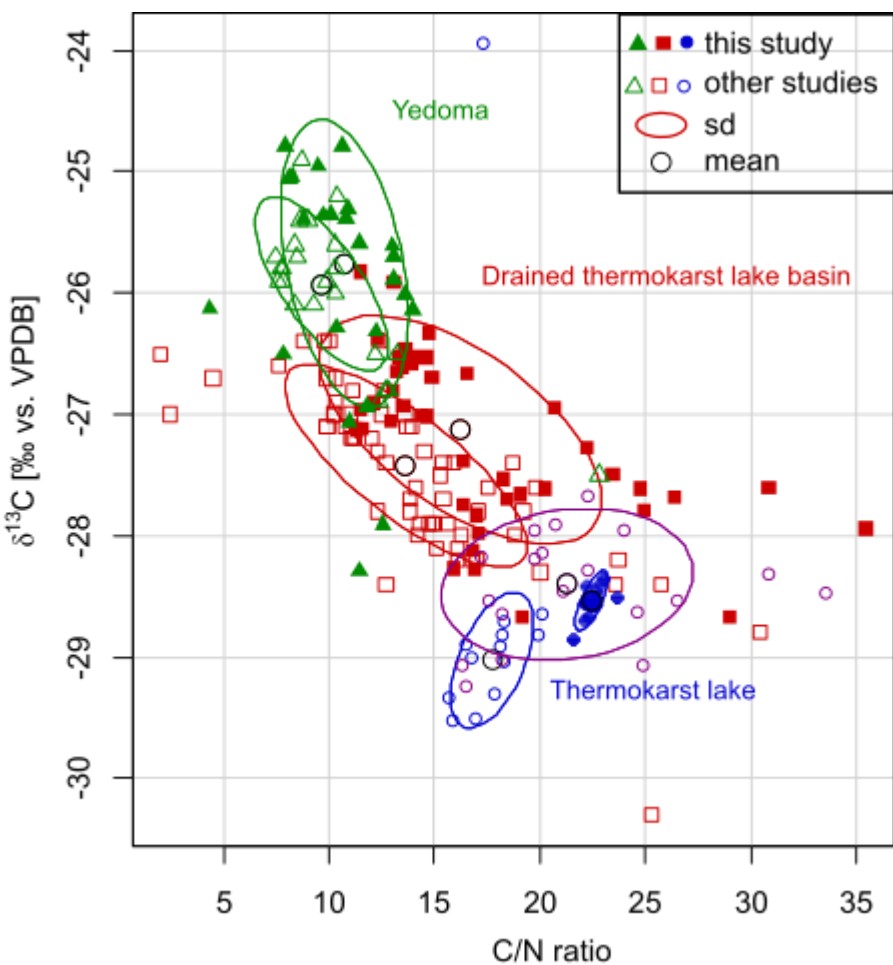

**Figure 8: Scatterplot of total organic carbon-total nitrogen (C/N) ratio and stable carbon isotopes (δ¹³C) of yedoma deposits (green triangles), drained thermokarst lake basin (DTLB) deposits (red squares) and thermokarst lake sediments (blue and purple dots) in Alaska. Mean value and standard deviation (sd) indicated. Data from this study (filled symbols): Baldwin Peninsula; data from other studies (hollow symbols): Itkillik River (Lapointe et al., 2017; Strauss et al., 2012), Northern Seward Peninsula (Lenz et al., 2016a) and lakes from Kobuk River Delta (blue) and Central Seward Peninsula (purple) (Lenz et al., 2018).**