# Peer review of "Organic matter characteristics in yedoma and thermokarst deposits on Baldwin Peninsula, West-Alaska"

_Biogeosciences, 2018_

## Referee Comment (RC1) · Anonymous Referee #1 · 1 May 2018

This manuscript is a characterization of different deposit types in a permafrost landscape in Western Alaska. The study is very sound and provides the necessary level of detail to be useful to the research community. The results from this study are a good addition to existing datasets and the authors do a nice job putting the results from Baldwin Peninsula in a larger context within the permafrost zone.

There were some confusing aspects in the manuscript that need to be addressed:

- A simple but crucial fix would be to use meaningful abbreviations for the different deposits throughout the manuscript (in all figures, tables, and text). I am sure the current naming system means something to the authors but it is very disruptive and

confusing to have to read BAL16-B2 and BAL16-UPL1-L1 for two different types of deposits. It should be possible to understand tables and figures without having to read the part about what the different labels mean. The flow of the manuscript would be much better with a simpler naming system.

- Section 3.1.4 Statistical significance: The statistics are too simplistic. I have to assume (you are not mentioning it or showing it) that your data are not normally distributed and that you therefore choose a non-parametric test, correct? The description of data distribution and statistical procedure is insufficient. Also, it is not appropriate to do pairwise comparisons when there are more than two groups without at least correcting for multiple testing. The minimum would be to perform a Kruskal-Wallis test and if significant to add a pairwise Wilcoxon-test, which would calculate pairwise comparisons between groups (you would have to include a correction for multiple testing). But, I think even that approach is too simplistic. You have multiple depths at different sites and so it does not make sense to compare one site with the other when you are not comparing the same thing. You could consider binning your data to different depths or different ages per site and then perform statistical analyses, preferably an ANOVA or something. The statistical results are not the core of your manuscript and that is fine, it still needs to be accurate.

- C/N already says it is a ratio and you do not need to add ratio afterwards

- The photographs in the Supplementary material are useful

- Table 1: I don't understand what Mean cal ages and rounded 14C ages are and why only a few samples had a +/-. You need to explain what +/- is

- Why are you showing 14C Ages again in Fig. 3? Isn't it the same as in Figure 2? It seems redundant but maybe there is a reason for it and an explanation is needed

- Fig. 2 and 3, it would be better to show dots for all the other variables as well and not just for the age column. You are only measuring a few data points along the profile

and it gives a slightly wrong impression if you show lines as you are not continuously measuring

- Fig. 3, why did you not measure biomarkers in the lake sediment? In the method section you say that you only measured it in those two but you don't say why

- When describing results along a depth gradient I think it is much better to go from the surface downwards and not the other way around. All soils have a surface but they go to a different depths and that just makes it confusing

- Section 3.1.3, add that the data to this section are shown in Figure 2 (bottom panels)

- Table S2, here you introduce new acronyms for Yedoma when previously you have used this awkward BAL16-B2 naming, I very strongly suggest that you use informative labels in all figures, tables, and throughout the text

- Figure 4: can probably be moved to the Supplementary Material. Is the number n-C29 or n-C31 that is indicated in the x-axis the dominant chain? I find this figure confusing.

- The grain size distribution figures in the supplementary material require more explanation. Should S7 be for BAL16-B4 and not B2? You show BAL16-B2 in S6. What is B2.1 through B2.42 in S6 and B4.1 through B4.31 in S7? You need to write out what f., m., c., v.c. means.

- Table S2, S3, and S4, what is "outcome of Mann-Whitney-Wilcoxon test'? P-values? Which software did you use? Please also use a consistent number of digits after the comma. I would refrain from adding stars to non-significant outcomes as that is usually used to indicate significance. The table would be so much easier to read if you had less numbers per cell, why not just indicate p-values as <0.05, <0.01, and <0.001 or something like that.

- Figure 6 could be moved to Supplementary Materials

- Discussion: the discussion is very hard to read because it often is a listing of results

followed by another listing of results from other publications. Some re-organization and focus on the important results would help the story line.

- I think it is useful that you compare the results from Baldwin Peninsula with previous studies, I am hesitant to believe the statistical results at this point because of my previous comments in regard to statistics

---

## Referee Comment (RC2) · Anonymous Referee #2 · 28 May 2018

Permafrost affected soils and sediments of the Northern hemisphere are a major terrestrial C reservoir, highly vulnerable to climate change. A better knowledge on the amount and composition of organic matter is thus crucial (e.g. to improve earth system models). Thus the authors report on a very important topic in biogeochemical research. However, the authors miss to get a clear central theme. It seems the group of authors tried to include a bit of everything in a very descriptive manner rather than providing a synthesis of the extensive data set. Another major drawback is the rather one-sided citation of studies either from the co-author list or affiliated colleagues. Especially with respect to organic matter quantity and quality a growing number of biogeochemical basic research is going on in the Arctic. For instance Gentsch et al. worked on the

bioavailability of specific OM in Siberia, or Mueller et al. worked rather "close by" on OM quantity and quality on thaw lake basins in the Alaskan North slope region.

See detailed remarks below:

line 21 Volumetric OC content in your case is OC stock. With giving soil OC stocks you are closer to what gets reported for soils.

page 3 line 6, To which OC pool do you refer here? Are you aiming to model specific OC pools with respect to decomposability, or are you just aiming to differentiate OC stocks with respect to different research sites?

page 3, line 7 The used biomarkers only represent a minor portion of the organic matter. Although useful for reconstruction of OM origin, these proxies are lower in explanatory power for the bioavailability of the sequestered OM. So I would not speak of "molecular composition of each OC pool and its quality" as it only represents a minor part of the bulk OC.

page 4, line 10-23 - What was the reason to go to this site? How representative is it for Arctic permafrost soil landscapes with respect to the studied OC distribution and composition?

page 3 line 28 - What do you mean by representative? How did you test representativity? How are the five locations connected to each other with respect to the choice of sampling spots?

page 5 line 2 - How were the samples pre-treated? Did the authors test for Carbonates in the samples, or is the TC representing OC and IC?

page 5 line 8 - Was it not possible to increase the sample amount to get into the measurement range?

page 5 line 15-28 - You are extracting free lipids, and thus you can make assumptions about the composition of the extractable lipid fraction of your samples. You can not

draw conclusions about the "molecular composition of the OC" in general as proposed. Please be more precise in the writing.

page 6/7 line 30 and following - You are taking some samples at one small edge of the Island and estimate based on this the OC stocks for the whole Island? Do you have any data on the representativity of the sampled locations for the rest of the Island? And what is the aim of such a very vague approximation? I miss a consequent central theme in the manuscript. Is it the quantification of OC stocks in a permafrost affected landscape? If yes, you clearly miss representivity (e.g. just one lake core!). Or is it the study of the composition of the extractable lipids in concert with C and N contents? If yes, you could possibly dig deeper into that by looking for correlations between all the measured data. Results - How are all the single proxies/data correlated? You are just reporting every single measured proxy, but how are things related to each other?

page 9 line 7 - What is the uncertainty based on the spatial heterogeneity of sediment and soil properties including BD, C content, horizon depths etc.? How did you account for the spatial heterogeneity on the Island with respect to only 5 sampling spots at the edge of the research area?

page 9 line 20 and following - What does this paragraph in its extensive form have to do with "organic carbon characteristics" as proposed in the title? I recomend to at least shorten the "origin of the material" section, or put very reduced parts of it into the site description in the M&M section. The parts with 14C and 13C etc. should go into a condensed discussion of the OM composition in the subsequent section.

page 11 line 2 - So if it is comparable, why should one keep on reading? Put your data first and get the central theme out of it, not just repeat other peoples work at a new fancy sampling location.

page 11 line 8 - "a significant OC pool is expected" - do you have data to prove it? Otherwise stay away from vaque approximations.

page 13 line 13-27 - This whole paragraph is purely hypothetical. You have no data on OC vulnerability to climate warming nor for OC bioavailability. What is the central theme of your work? It reads like the authors wanted to have a bit of everything in it, paleo reconstruction, large scale OC estimates and OC composition. It would be great to get a synthesis of these parts rather than a descriptive manuscript.

Conclusions – This is just a summary of your findings, but what are the take home messages and especially the implications of your work?

---

## Author Response (AR1)

Authors reply by Loeka Jongejans, on behalf of all authors (loeka.jongejans@awi.de)

We thank this referee for the positive feedback on our manuscript and their constructive comments. We will revise our manuscript according to your suggestions. Below, we addressed and replied to all the suggestions and questions that were raised.

Referee Comment (RC): This manuscript is a characterization of different deposit types in a permafrost landscape in Western Alaska. The study is very sound and provides the necessary level of detail to be useful to the research community. The results from this study are a good addition to existing datasets and the authors do a nice job putting the results from Baldwin Peninsula in a larger context within the permafrost zone.

Authors Reply (AR): We thank the referee for their positive feedback and for acknowledging the importance of our study.

RC: A simple but crucial fix would be to use meaningful abbreviations for the different deposits throughout the manuscript (in all figures, tables, and text). I am sure the current naming system means something to the authors but it is very disruptive and confusing to have to read BAL16-B2 and BAL16-UPL1-L1 for two different types of deposits. It should be possible to understand tables and figures without having to read the part about what the different labels mean. The flow of the manuscript would be much better with a simpler naming system.

AR: We thank the referee for their suggestion. We decided to keep the names of the study sites, as these are the names given to the samples during the field work and laboratory analyses and it would complicate data management and overview when we would rename the samples. Following the suggestion of the reviewers, we explain the naming system by adding Table 1 (see below) with the study sites with corresponding names in chapter 2.3. We think that a consistent and defined naming system improves the reading flow as then a stratigraphical orientation is clear by reading the sample/site name.

Table 1: Overview of study sites of yedoma exposure, drained thermokarst lake basin (DTLB) exposure and thermokarst lake sediments including coordinates and number of samples. BAL: Baldwin Peninsula, 16: year of expedition, 2016, B: bluff sampling, UPL: upland sampling, L: lake sampling.

| Study site ID | Landscape unit | Coordinates | Samples, n |
|---|---|---|---|
| BAL16-B2 | Yedoma exposure | 66.73262°N, 62.49450°W | 18 |
| BAL16-B4 | DTLB exposure | 66.73644°N, 62.50208°W | 31 |
| BAL16-UPL1-L1 | Thermokarst lake sediments | 66.74220°N, 62.41310°W | 9 |

RC: Section 3.1.4 Statistical significance: The statistics are too simplistic. I have to assume (you are not mentioning it or showing it) that your data are not normally distributed and that you therefore choose a non-parametric test, correct? The description of data distribution and statistical procedure is insufficient. Also, it is not appropriate to do pairwise comparisons when there are more than two groups without at least correcting for multiple testing. The minimum would be to perform a Kruskal-Wallis test and if significant to add a pairwise Wilcoxon-test, which would calculate pairwise comparisons between groups (you would have to include a correction for multiple testing). But, I think even that approach is too simplistic. You have multiple depths at different sites and so it does not make sense to compare one site with the other when you are not comparing the same thing. You could consider binning your data to different depths or different ages per site and then perform statistical analyses, preferably an ANOVA or something. The statistical results are not the core of your manuscript and that is fine, it still needs to be accurate.

AR: Thank you for the suggestion. We improved our explanation on the applied statistics in the text and the supplement. We now also performed a normality test. However, non-parametric tests also work for normally distributed data. We see some conflicts with the aim of our study concerning the binning of our data. Our goal is to compare the different parameters between the different landscape

units in order to see the range and variation of the data per landscape unit. Therefore, we do not bin the data to different depths but take the landscape unit as a whole. We explained this better in the revised manuscript. We added the results of the Kruskal-Wallis tests to the supplementary information.

RC: C/N already says it is a ratio and you do not need to add ratio afterwards

AR: We changed this throughout the manuscript. Thank you for the suggestion.

RC: The photographs in the Supplementary material are useful

AR: Thank you

RC: Table 1: I don't understand what Mean cal ages and rounded 14C ages are and why only a few samples had a +/-. You need to explain what +/- is

AR: The Mean cal ages (changed to Calibrated ages in the revised manuscript) are the mean values of the age range that were derived from the 14C calibration software (using CALIB 7.1 and IntCal13 calibration curve). The calibrated ages are shown including one standard deviation ($\sigma$) uncertainty ($\pm$). The calibration is not possible for samples with infinite ages (>50,000 years) and therefore, no uncertainty is given for these samples. We clarified this in the revised version of our manuscript.

RC: Why are you showing 14C Ages again in Fig. 3? Isn't it the same as in Figure 2? It seems redundant but maybe there is a reason for it and an explanation is needed

AR: The reviewer is right; the 14C Ages in Fig. 3 are indeed the same as in Fig. 2, which we clarified in the figure caption. We chose to show the ages again in Fig. 3 as the timeline inferred from them support the discussion on the biomarkers present in the sediments.

RC: Fig. 2 and 3, it would be better to show dots for all the other variables as well and not just for the age column. You are only measuring a few data points along the profile and it gives a slightly wrong impression if you show lines as you are not continuously measuring

AR: We thank the reviewer for their comment and revised the graphs accordingly.

RC: Fig. 3, why did you not measure biomarkers in the lake sediment? In the method section you say that you only measured it in those two but you don't say why

AR: The initial focus of the project was on the terrestrial deposits and therefore the biomarker analysis is too. We later decided to also include lacustrine sediments to cover all three main landscape units of the Baldwin Peninsula. However, only general biogeochemical properties were analyzed for this landscape type.

RC: When describing results along a depth gradient I think it is much better to go from the surface downwards and not the other way around. All soils have a surface but they go to a different depths and that just makes it confusing

AR: We agree with the reviewer that soils are generally formed from the surface downwards and that age increases with depth. However, the permafrost deposits in this study, and the yedoma in particular, have built up with time. Therefore, we decided to follow the geological timeframe, also considering that the time of deposition has a big influence on the studied sediments.

RC: Section 3.1.3, add that the data to this section are shown in Figure 2 (bottom panels)

AR: We added this reference as suggested.

RC: Table S2, here you introduce new acronyms for Yedoma when previously you have used this awkward BAL16-B2 naming, I very strongly suggest that you use informative labels in all figures, tables, and throughout the text

AR: We changed the names in Table S2 and S3 to match the names used throughout the manuscript.

RC: Figure 4: can probably be moved to the Supplementary Material. Is the number n-C29 or n-C31 that is indicated in the x-axis the dominant chain? I find this figure confusing.

AR: The figure was moved to the Supplements as suggested. The number below the x-axis is indeed the dominant chain per sample, which we clarified in the revised manuscript.

RC: The grain size distribution figures in the supplementary material require more explanation in the revised manuscript. Should S7 be for BAL16-B4 and not B2? You show BAL16-B2 in S6. What is B2.1 through B2.42 in S6 and B4.1 through B4.31 in S7? You need to write out what f., m., c., v.c. means.

AR: The first graph in the section Grain size distribution (previously S7, S8 in the revised version) shows the grain size distribution of the yedoma exposure, the second graph (previously S8, now S9) the drained thermokarst lake basin. Numbers B2.1 through B2.42 stand for the subsamples of the exposure. In the revised manuscript, we indicated the depths instead. The same changes were made for the drained thermokarst lake basin graph. The letters f., m., etc. refer to different grain size classes, which we explained in the revised manuscript as suggested.

RC: Table S2, S3, and S4, what is "outcome of Mann-Whitney-Wilcoxon test'? P-values? Which software did you use? Please also use a consistent number of digits after the comma. I would refrain from adding stars to non-significant outcomes as that is usually used to indicate significance. The table would be so much easier to read if you had less numbers per cell, why not just indicate p-values as <0.05, <0.01, and <0.001 or something like that.

AR: We calculated the Kruskal-Wallis-Test and Mann-Whitney-Wilcoxon using R (version 3.4.3) (kruskal.test and wilcox.test, respectively). To improve clarity, we changed the notation of the p-values in Table 2, 3 and 4 as suggested (<0.05, <0.01 and <0.001).

RC: Figure 6 could be moved to Supplementary Materials

AR: As suggested, we moved Figure 6 to the Supplements and we added the references to the text.

RC: Discussion: the discussion is very hard to read because it often is a listing of results followed by another listing of results from other publications. Some re-organization and focus on the important results would help the story line.

AR: Thank you for the comment. We restructured the discussion as suggested and shifted the focus to comparing our data to those of other studies.

RC: I think it is useful that you compare the results from Baldwin Peninsula with previous studies, I am hesitant to believe the statistical results at this point because of my previous comments in regard to statistics

AR: In order to identify the differences or similarities between the landscape units between this and other studies in Alaska. We performed pairwise comparisons (Mann-Whitney-Wilcoxon test) and used these results in the discussion.

Authors reply by Loeka Jongejans, on behalf of all authors (loeka.jongejans@awi.de)

We thank this referee for the valuable feedback on our manuscript. We went through the discussion section and incorporated their suggestions. We hope to clarify the central theme of our manuscript in our replies to their open questions.

Referee Comment (RC): Permafrost affected soils and sediments of the Northern hemisphere are a major terrestrial C reservoir, highly vulnerable to climate change. A better knowledge on the amount and composition of organic matter is thus crucial (e.g. to improve earth system models). Thus the authors report on a very important topic in biogeochemical research. However, the authors miss to get a clear central theme. It seems the group of authors tried to include a bit of everything in a very descriptive manner rather than providing a synthesis of the extensive data set. Another major drawback is the rather one-sided citation of studies either from the co-author list or affiliated colleagues. Especially with respect to organic matter quantity and quality a growing number of biogeochemical basic research is going on in the Arctic. For instance Gentsch et al. worked on the bioavailability of specific OM in Siberia, or Mueller et al. worked rather "close by" on OM quantity and quality on thaw lake basins in the Alaskan North slope region.

Authors Reply (AR): In our study, we aim to characterize the OC properties in permafrost deposits in order to assess the vulnerability of the permafrost to climate change and contribute to a better estimate of the terrestrial C reservoir in this part of the Arctic. We highlighted this theme better in the revised version of the manuscript. Regarding the comment on one-sided citation, in particular the OC quantity, we compared our OC budget estimates to other studies that studied similar deposits (yedoma or DTLB) and were expressed in the same units (kg/m3). In the revised manuscript we added the suggested study of Mueller et al. (2015). Regarding the OC quality, we elaborated on bioavailability of OC in high-latitude soils and included more studies such as the suggested study of Gentsch et al. (2015), and Vonk et al. (2010).

RC: line 21 Volumetric OC content in your case is OC stock. With giving soil OC stocks you are closer to what gets reported for soils.

AR: With volumetric OC content we mean the OC density, as the values are expressed per unit weight over unit volume. The carbon stock, however, is expressed per unit weight, which we report in megaton (Mt).

RC: page 3 line 6, To which OC pool do you refer here? Are you aiming to model specific OC pools with respect to decomposability, or are you just aiming to differentiate OC stocks with respect to different research sites?

AR: We here refer to the OC pool in the yedoma, DTLB and thermokarst lake sediments. We aim to estimate the stocks and decomposability, i.e. size and quality of OC pools. We clarified this in the revised version of the manuscript.

RC: page 3, line 7 The used biomarkers only represent a minor portion of the organic matter. Although useful for reconstruction of OM origin, these proxies are lower in explanatory power for the bioavailability of the sequestered OM. So I would not speak of "molecular composition of each OC pool and its quality" as it only represents a minor part of the bulk OC.

AR: We agree with the reviewer and rephrased this throughout the manuscript.

RC: page 3, line 10-23 - What was the reason to go to this site? How representative is it for Arctic permafrost soil landscapes with respect to the studied OC distribution and composition?

AR: This is the first time yedoma deposits on Baldwin Peninsula were described. Therefore, this study contributes to a better and more precise approximation of the OC of yedoma deposits for this part of the Arctic. We made this clearer in the revised version of the manuscript. The yedoma deposits were

discovered on the coast of the peninsula during a reconnaissance campaign, after which this coastal bluff and those of the drained thermokarst lake basin deposits were sampled. We show that the total organic carbon content of the yedoma deposits is in the range of other yedoma studies in Siberia and Alaska. Also, the higher quality of yedoma OC compared to that in DTLB deposits was shown before. We address the representativity of our study site in the outlook, the past paragraph of the discussion.

RC: page 3 line 28 - What do you mean by representative? How did you test representativity? How are the five locations connected to each other with respect to the choice of sampling spots?

AR: With representative, we mean that we tried to cover the whole yedoma exposure. Due to the difficult terrain and the fact that the samples were taken in summer – the fast thawing of the deposits limits the accessibility – it was not possible to sample the whole exposure in one straight profile. Therefore, we sampled different portions of the exposure wherever accessible and compiled a composite profile. We changed this explanation in the manuscript.

RC: page 5 line 2 - How were the samples pre-treated? Did the authors test for Carbonates in the samples, or is the TC representing OC and IC?

AR: We did not pre-treat the samples. We measure the TC and TOC in different devices. To measure the TOC, the samples are combusted at a much lower temperature compared to the TC measurements, so that the inorganic part of the sample is not combusted, and hence not measured in the device. The total carbon (TC) represents the sum of the organic (TOC) and inorganic carbon (TIC).

RC: page 5 line 8 - Was it not possible to increase the sample amount to get into the measurement range?

AR: The sensitivity of the Elementar Vario Max C is 0.1 wt%. This means that with a large sample amount, the weight percentage would be similar and therefore also below the detection limit of the device. We measured two aliquots per sample where we allowed a standard deviation of <5%; we measured multiple times when this criterion was not met.

RC: page 5 line 15-28 - You are extracting free lipids, and thus you can make assumptions about the composition of the extractable lipid fraction of your samples. You cannot draw conclusions about the "molecular composition of the OC" in general as proposed. Please be more precise in the writing.

AR: You are right, we rephrased as suggested.

RC: page 6/7 line 30 and following - You are taking some samples at one small edge of the Island and estimate based on this the OC stocks for the whole Island? Do you have any data on the representativity of the sampled locations for the rest of the Island? And what is the aim of such a very vague approximation? I miss a consequent central theme in the manuscript. Is it the quantification of OC stocks in a permafrost affected landscape? If yes, you clearly miss representivity (e.g. just one lake core!). Or is it the study of the composition of the extractable lipids in concert with C and N contents? If yes, you could possibly dig deeper into that by looking for correlations between all the measured data.

AR: Arctic fieldwork is expensive and it is difficult to get to the remote places for sampling. Therefore, we sampled the three main landscape units of the peninsula to get an initial overview of the thermokarst processes influencing the topography and the organic carbon characteristics. The sample sites are exposed at the coast, allowing us to study OC characteristics of deep permafrost deposits. Sampling sites on top of the deposits, however, would require drilling. This was not possible as no drilling rig was available during the fieldwork. Using the stratigraphical land cover classification map that we made and remote sensing, we indeed generated a first estimate of the characteristics and size of the OC pool in this part of the Arctic.

RC: Results - How are all the single proxies/data correlated? You are just reporting every single measured proxy, but how are things related to each other?

AR: Our aim is to characterize the OC pool in the different landscape units on Baldwin Peninsula by assessing the OC pool size and quality. In order to assess the organic carbon quantity, we analyzed the total organic carbon content and – using the stratigraphical landcover classification map we made and bootstrapping techniques – we estimated the OC stock of the different landscape units (based on the wedge-ice volume, bulk density, total organic carbon content and the coverage and thickness of the deposits). In order to assess the OC decomposability, i.e. the quality, we analyzed the carbon-nitrogen ratio and stable carbon isotopes. Using the differences between the landscape units, we show that the C/N ratio and d13C show both OC source as well as quality. We used the n-alkane derived ACL index to distinguish between terrestrial land plants, algae and bacteria, and the CPI index as an indicator for OC decomposability, where a higher CPI suggests well preserved material. We clarified the link between the parameters in the revised manuscript. However, we would like to keep the results section as it is to keep it factual.

RC: page 9 line 7 - What is the uncertainty based on the spatial heterogeneity of sediment and soil properties including BD, C content, horizon depths etc.? How did you account for the spatial heterogeneity on the Island with respect to only 5 sampling spots at the edge of the research area?

AR: This study represents the first characterization of yedoma deposits on Baldwin Peninsula. Even though spatial heterogeneity exists both between and within landscape units (e.g. Zona et al., 2011), we were able collect a total of 91 samples at 5 different locations that we used for all analyses. We believe that this is sufficient for an initial characterization of the OC pool in this part of the Arctic, and the objective of our study. The uncertainties of the estimations are included by repeated artificial subsampling for the OC stock calculation using bootstrapping. A detailed assessment of potential spatial heterogeneity is beyond the scope of this paper. However, following the suggestions of the reviewers, we sharpened the existing focus of the paper in the revised manuscript.

RC: page 9 line 20 and following - What does this paragraph in its extensive form have to do with "organic carbon characteristics" as proposed in the title? I recomend to at least shorten the "origin of the material" section, or put very reduced parts of it into the site description in the M&M section. The parts with 14C and 13C etc. should go into a condensed discussion of the OM composition in the subsequent section.

AR: This section describes the depositional environment of the island to provide a framework for the interpretation of the OC data. In order to assess the vulnerability of carbon in permafrost deposits on Baldwin Peninsula, it is crucial to know the source and properties of the deposits in which this OC is stored. Especially for OC in old, deep permafrost deposits like yedoma, it is highly important to report on sedimentary origin as well as the ages of the deposits. These both describe the regional geological context of the deposits, as well as it gives an insight in the origin of the material. Hence, we prefer to keep the content of this paragraph as it is.

RC: page 11 line 2 - So if it is comparable, why should one keep on reading? Put your data first and get the central theme out of it, not just repeat other peoples work at a new fancy sampling location.

AR: Thank you for the suggestion. We restructured the paragraph.

RC: page 11 line 8 - "a significant OC pool is expected" - do you have data to prove it? Otherwise stay away from vaque approximations.

AR: We wanted to stress the importance of the volume of the yedoma deposits compared to the relatively shallow thermokarst deposits, after which we report on the absolute numbers. To avoid further confusion, we rephrased this sentence as suggested.

RC: page 13 line 13-27 - This whole paragraph is purely hypothetical. You have no data on OC vulnerability to climate warming nor for OC bioavailability. What is the central theme of your work? It reads like the authors wanted to have a bit of everything in it, paleo reconstruction, large scale OC estimates and OC composition. It would be great to get a synthesis of these parts rather than a descriptive manuscript.

AR: Giving a synthesis of the OC pool size and composition in permafrost on Baldwin Peninsula is exactly what we tried to do in the first part of the discussion, where we report on the quantity and quality of OC based on organic geochemical, sedimentological and also palaeoecological methods. The paragraph following this is meant as an outlook and to put the study and also the study area in a larger perspective. We put more stress on the actual data rather than on the more hypothetical part in the revised version of our manuscript.

RC: Conclusions – This is just a summary of your findings, but what are the take home messages and especially the implications of your work?

AR: We agree with the reviewer: we indeed summarized our findings in the conclusion. With these main findings, we show the answer to our research question and our main message: the first estimate of the total OC pool on Baldwin Peninsula, the relative contribution of the different landscape units (answer to our first research question), as well as the finding that OC in yedoma is most vulnerable to decomposition (answer to our second research question).

[revised manuscript text omitted]

**2.2** **Total organic carbon**

Figure S6 shows variations in total organic carbon of previously studied yedoma deposits in Siberia and Alaska.

[Figure]

5    **Figure S6: Total organic carbon (TOC) variations from different yedoma study sites in Siberia and Alaska. Sorted from westernmost (Cape Mamontov Klyk, western Laptev Sea) to easternmost (Itkillik River outcrop, Alaskan North Slope) study sites. Data from Schirrmeister et al. (2008a, 2008b, 2011), Strauss et al. (2012, 2013, 2015) and Baldwin Peninsula (this study; blue).**

**2.3 Additional profiles**

Figure S7 shows the cryostratigraphical and biogeochemical parameters of the additional drained thermokarst lake basin exposures BAL16-B3 and BAL16-B5 that were used in the organic carbon calculations.

[Figure]

Figure S7: Summary of cryostratigraphical and biogeochemical parameters of BAL16-B3 and BAL16-B5 (drained thermokarst lake basin exposures): bulk density, total organic carbon (TOC), total organic carbon-total nitrogen (C/N) ratio (C/N), stable carbon isotopes (δ13C).

**2.4  *n*-Alkane concentration**

The *n*-alkane concentrations per sample are shown in Figure S8. Also, the dominating *n*-C chain is indicated.

[Figure]

Figure S8: *n*-alkane concentrations of BAL16-B2 (yedoma) and BAL16-B4 (drained thermokarst lake basin) by depth. Odd chains (grey bars) and even chains (red bars), sample depth (above graph) and dominating *n*-chain indicated (below x-axis). Note: different y-axes.

 **Depositional environment**

         **Grain size distribution**

The grain size distributions of yedoma exposure BAL16-B2 and drained thermokarst lake basin exposure BAL16-B4 are shown in Figure S9 and Figure S10, respectively.

[Figure]

[Figure]

**Figure S9: Grain size distribution of**  **BAL16-B2** (vedoma). **Samples sorted over depth (legend on the right). v.f.: very fine, f.: fine, m.: medium, c.: coarse, v.c.: very coarse.** Sediments from the separate unit underlying the yedoma deposits indicated.

[Figure]

**Figure S10: Grain size distribution of**  **BAL16-B4** (drained thermokarst lake basin). **Samples sorted over depth (legend on the right). v.f.: very fine, f.: fine, m.: medium, c.: coarse, v.c.: very coarse.** Sediments from depth interval of 280-385 cm indicated.

**2.4.22.5.2 Climatic indicators**

Table S1 shows the brGDGT derived climatic indices BIT and MBT indices.

**Table S1: brGDGT derived climatic indices for  BAL16-B2 (yedoma) and  BAL16-B4 (drained thermokarst lake basin): branched and isoprenoid tetraethers (BIT) index and methylation of branched tetraethers (MBT) index.**

| | Depth [cm] | BIT | MBT |
|---|---|---|---|
| | 620 | 0.90 | 0.19 |
| | 945 | 0.96 | 0.19 |
| | 1117 | NA | NA |
| Yedoma | 1399 | 1.00 | 0.39 |
| | 1500 | 0.97 | 0.20 |
| | 1600 | 0.88 | 0.18 |
| | 8 | 1.00 | 0.25 |
| | 120 | 1.00 | 0.27 |
| | 160 | 1.00 | 0.30 |
| DTLB | 340 | 0.99 | 0.25 |
| | 430 | 1.00 | 0.22 |
| | 583 | 1.00 | 0.16 |
| | 798 | NA | NA |

**2.5 2.6   Statistical tests**

Using the Shapiro-Wilk test, we tested for normality of the data (Table S2). In this test, the null hypothesis states that the data are normally distributed. When the p-value exceeds 0.05, the null hypothesis cannot be rejected. Using Mann-Whitney-Wilcoxon the Kruskal-Wallis test, a non-parametric test, we tested for significant differences of the biogeochemical parameters (bulk density, total organic carbon, carbon-nitrogen ratio and stable carbon isotopes) between the three stratigraphic landscape units on Baldwin Peninsula (Table S2): yedoma, drained thermokarst lake basin and thermokarst lake sediments. In this test, the null hypothesis states that there is a no statistically significant difference between the samples of the different landscape units. When the p-value exceeds 0.05, the null hypothesis cannot be rejected. A Mann-Whitney-Wilcox test was added for pairwise comparisons between the landscape units (Table S2).

The Mann-Whitney-Wilcoxon test was also used to test for significant differences based on the C/N ratio (Table S3) and the δ[13]C value (Table S4) between this study (Baldwin Peninsula) and other studies from Alaska: yedoma deposits along the Itkillik River (IR) (Lapointe et al., 2017; Strauss et al., 2012), DTLB deposits from the Northern Seward Peninsula (NSP) (Lenz et al., 2016) and thermokarst lake sediments from lakes in the Kobuk River Delta (KOB) and Central Seward Peninsula (CSP) (Lenz et al., 2018).

**Table S2:** Outcome statistical tests Mann-Whitney-Wilcoxon test of bulk density (BD), total organic carbon (TOC), carbon-nitrogen (C/N) ratio and stable carbon isotopes (δ[13]C) between yedoma(Y) (BAL16-B2 (yedoma)), BAL16-B4 (drained thermokarst lake basin; (DTLB) (BAL16-B4) and BAL16-UPL1-L1 (thermokarst lake) (TL) (BAL16-UPL1-L1) sediments on Baldwin Peninsula.

[revised manuscript text omitted]

---

## Author Response (AR2)

Authors reply by Loeka Jongejans, on behalf of all authors (loeka.jongejans@awi.de)

We thank the referee for their constructive comments. We have revised our manuscript accordingly to your suggestions. Below, we addressed and replied to the suggestions that you made.

Referee Comment (RC): I completely agree with the authors that especially in remote arctic environments it is often not possible to use sampling schemes with a high number of replicates. And every data set we can get from these areas is worth the effort. Nevertheless, the issue of low representativity has to taken into account when discussing data or setting it into a larger context, for instance with respect to a whole Island.

Authors Reply (AR): Our work provides a first estimate of the OC pool size of the Baldwin Peninsula deposits. It is based on the data from the five study sites which correspond to the main landscape units in the study area: yedoma (30% of surface area), drained thermokarst lake basin (65%) and thermokarst lake deposits (5%). Furthermore, these three landscape units make up the largest share of the total yedoma permafrost region, where generally only a small part is covered by other sediments such as deltaic and fluvial sediments (Jorgenson et al., 2008; Romanovskii, 1993; Strauss et al., 2013). For this estimate we assumed homogeneity over the Baldwin Peninsula. In addition, bootstrapping the results helps constrain the standard error of the estimate. We are certain that further studies will help provide a better constrain on our estimate as more samples are added and spatial heterogeneity is addressed.

RC: With respect to the "volumetric OC" - unit weight over unit volume (which is also defined by an area e.g. sqm) is exactly OC stock, so here we seem to just have a case of different wording between soil science and paleo/geography.

AR: Thank you for the clarification.

RC: With respect to a phrase in the abstract: "importance of lipid biomarker analysis for determining the potential future greenhouse gas emissions from thawing permafrost" - here you definitely overrating lipid biomarkers. You should re-phrase these statements.
This also applies to the phrase "high quality" with respect to the lipid biomarkers. What is "high quality"? I guess you refer here to OM that has still the signature of the initial source material. I would be more specific here. For instance some studies refer to "low quality" OM with respect to sawdust or straw, materials that are definitely comparable to fresh arctic dwarf shrub or sedge vegetation. I would recommend being more specific here and refer to "unaltered OM" or comparable phrases.

AR: We thank you for the comment and rephrased the sentence in the abstract as suggested.
Indeed, quality refers to OM decomposability and high quality refers to well preserved material. We added a more detailed explanation in the introduction and made appropriate changes throughout the manuscript.

RC: With respect to the Conclusions I still think you could do better in terms of carving out the implications of your work, rather than just reporting your data.

AR: We made changes to the conclusion and added a sentence highlighting the implications of this study.

[revised manuscript text omitted]

**2.2 Total organic carbon**

Figure S6 shows variations in total organic carbon of previously studied yedoma deposits in Siberia and Alaska.

[Figure]

**Figure S6: Total organic carbon (TOC) variations from different yedoma study sites in Siberia and Alaska. Sorted from westernmost (Cape Mamontov Klyk, western Laptev Sea) to easternmost (Itkillik River outcrop, Alaskan North Slope) study sites. Data from Schirrmeister et al. (2008a, 2008b, 2011), Strauss et al. (2012, 2013, 2015) and Baldwin Peninsula (this study; blue).**

**2.3 Additional profiles**

Figure S7 shows the cryostratigraphical and biogeochemical parameters of the additional drained thermokarst lake basin exposures BAL16-B3 and BAL16-B5 that were used in the organic carbon calculations.

[Figure]

**Figure S7: Summary of cryostratigraphical and biogeochemical parameters of BAL16-B3 and BAL16-B5 (drained thermokarst lake basin exposures): bulk density, total organic carbon (TOC), total organic carbon-total nitrogen ratio (C/N), stable carbon isotopes ($\delta^{13}$C).**

**2.4 *n*-Alkane concentration**

The *n*-alkane concentrations per sample are shown in Figure S8. Also, the dominating *n*-C chain is indicated.

[Figure]

**Figure S8:** *n*-alkane concentrations of BAL16-B2 (yedoma) and BAL16-B4 (drained thermokarst lake basin) by depth. Odd chains (grey bars) and even chains (red bars), sample depth (above graph) and dominating *n*-chain indicated (below x-axis). Note: different y-axes.

**2.5 Depositional environment**

**2.5.1 Grain size distribution**

The grain size distributions of yedoma exposure BAL16-B2 and drained thermokarst lake basin exposure BAL16-B4 are shown in Figure S9 and Figure S10, respectively.

[Figure]

**Figure S9: Grain size distribution of BAL16-B2 (yedoma). Samples sorted over depth (legend on the right). v.f.: very fine, f.: fine, m.: medium, c.: coarse, v.c.: very coarse. Sediments from the separate unit underlying the yedoma deposits indicated.**

[Figure]

**Figure S10: Grain size distribution of BAL16-B4 (drained thermokarst lake basin). Samples sorted over depth (legend on the right). v.f.: very fine, f.: fine, m.: medium, c.: coarse, v.c.: very coarse. Sediments from depth interval of 280-385 cm indicated.**

**2.5.2 Climatic indicators**

Table S1 shows the brGDGT derived climatic indices BIT and MBT indices.

**Table S1: brGDGT derived climatic indices for BAL16-B2 (yedoma) and BAL16-B4 (drained thermokarst lake basin): branched and isoprenoid tetraethers (BIT) index and methylation of branched tetraethers (MBT) index.**

|  | Depth [cm] | BIT | MBT |
|---|---|---|---|
| Yedoma | 620 | 0.90 | 0.19 |
|  | 945 | 0.96 | 0.19 |
|  | 1117 | NA | NA |
|  | 1399 | 1.00 | 0.39 |
|  | 1500 | 0.97 | 0.20 |
|  | 1600 | 0.88 | 0.18 |
| DTLB | 8 | 1.00 | 0.25 |
|  | 120 | 1.00 | 0.27 |
|  | 160 | 1.00 | 0.30 |
|  | 340 | 0.99 | 0.25 |
|  | 430 | 1.00 | 0.22 |
|  | 583 | 1.00 | 0.16 |
|  | 798 | NA | NA |

**2.6 Statistical tests**

Using the Shapiro-Wilk test, we tested for normality of the data (Table S2). In this test, the null hypothesis states that the data are normally distributed. When the p-value exceeds 0.05, the null hypothesis cannot be rejected. Using the Kruskal-Wallis test, a non-parametric test, we tested for significant differences of the biogeochemical parameters (bulk density, total organic
5   carbon, carbon-nitrogen ratio and stable carbon isotopes) between the three stratigraphic landscape units on Baldwin Peninsula (Table S2): yedoma, drained thermokarst lake basin and thermokarst lake sediments. In this test, the null hypothesis states that there is no statistically significant difference between the samples of the different landscape units. A Mann-Whitney-Wilcox test was added for pairwise comparisons between the landscape units (Table S2).

The Mann-Whitney-Wilcoxon test was also used to test for significant differences based on the C/N (**Table S3**) and the $\delta^{13}C$
10   value (Table S4) between this study (Baldwin Peninsula) and other studies from Alaska: yedoma deposits along the Itkillik River (IR) (Lapointe et al., 2017; Strauss et al., 2012), DTLB deposits from the Northern Seward Peninsula (NSP) (Lenz et al., 2016) and thermokarst lake sediments from lakes in the Kobuk River Delta (KOB) and Central Seward Peninsula (CSP) (Lenz et al., 2018).

15   **Table S2: Outcome statistical tests of bulk density (BD), total organic carbon (TOC), carbon-nitrogen (C/N) ratio and stable carbon isotopes ($\delta^{13}C$) between BAL16-B2 (yedoma), BAL16-B4 (drained thermokarst lake basin; DTLB) and BAL16-UPL1-L1 (thermokarst lake) on Baldwin Peninsula.**

| P-value | | BD | TOC | C/N | $\delta^{13}C$ |
|---|---|---|---|---|---|
| Shapiro-Wilk Test | | <0.05 | <0.001 | <0.01 | <0.001 |
| Kruskal Wallis Test | | <0.05 | <0.001 | <0.05 | <0.001 |
| Wilcoxon test | Yedoma – DTLB | <0.05 | <0.001 | <0.001 | <0.001 |
| | Yedoma – Thermokarst lake | <0.001 | <0.001 | <0.001 | <0.001 |
| | DTLB – Thermokarst lake | <0.001 | <0.001 | <0.001 | <0.001 |

Table S3: Outcome Mann-Whitney-Wilcoxon test of C/N ratio between yedoma, drained thermokarst lake basin (DTLB) and thermokarst lake sediments. BP: Baldwin Peninsula (this study), IT: Itkillik River (Strauss et al., 2012), NSP: Northern Seward Peninsula (Lenz et al., 2016), KOB: Kobuk River Delta and CSP: Central Seward Peninsula (Lenz et al., 2018).

| P-value C/N | DTLB BAL16-B4 | Thermokarst lake BAL16-UPL1-L1 | Yedoma Itkillik River | DTLB Northern Seward Peninsula | Thermokarst lake Kobuk Delta | Thermokarst lake Central Seward Peninsula |
|---|---|---|---|---|---|---|
| Yedoma BAL16-B2 | <0.001 | <0.001 | >0.05 | <0.001 | <0.001 | <0.001 |
| DTLB BAL16-B4 | | <0.01 | <0.001 | <0.01 | >0.05 | <0.001 |
| Thermokarst lake BAL16-UPL1-L1 | | | <0.001 | <0.001 | <0.001 | >0.05 |
| Yedoma Itkillik River | | | | <0.001 | <0.001 | <0.001 |
| DTLB Northern Seward Peninsula | | | | | <0.001 | <0.001 |
| Thermokarst lake Kobuk Peninsula | | | | | | <0.01 |

**Table S4: Outcome Mann-Whitney-Wilcoxon test of $\delta^{13}C$ between yedoma, drained thermokarst lake basin (DTLB) and thermokarst lake sediments. BP: Baldwin Peninsula (this study), IT: Itkillik River (Strauss et al., 2012), NSP: Northern Seward Peninsula (Lenz et al., 2016), KOB: Kobuk River Delta and CSP: Central Seward Peninsula (Lenz et al., 2018).**

| P-value $\delta^{13}C$ | DTLB BAL16-B4 | Thermokarst lake BAL16-UPL1-L1 | Yedoma Itkillik River | DTLB Northern Seward Peninsula | Thermokarst lake Kobuk Delta | Thermokarst lake Central Seward Peninsula |
|---|---|---|---|---|---|---|
| Yedoma BAL16-B2 | <0.001 | <0.001 | >0.05 | <0.001 | <0.001 | <0.001 |
| DTLB BAL16-B4 | | <0.001 | <0.001 | >0.05 | <0.001 | <0.001 |
| Thermokarst lake BAL16-UPL1-L2 | | | <0.001 | <0.001 | <0.01 | >0.05 |
| Yedoma Itkillik River | | | | <0.001 | <0.001 | <0.001 |
| DTLB Northern Seward Peninsula | | | | | <0.001 | <0.001 |
| Thermokarst lake Kobuk Delta | | | | | | <0.01 |

**3 Supporting references**

Blott, S. J. and Pye, K.: GRADISTAT: a grain size distribution and statistics package for the analysis of unconsolidated sediments, Earth Surf. Process. Landf., 26(11), 1237–1248, doi:10.1002/esp.261, 2001.

Dirghangi, S. S., Pagani, M., Hren, M. T. and Tipple, B. J.: Distribution of glycerol dialkyl glycerol tetraethers in soils from two environmental transects in the USA, Org. Geochem., 59, 49–60, doi:https://doi.org/10.1016/j.orggeochem.2013.03.009, 2013.

Hopmans, E. C., Weijers, J. W. H., Schefuß, E., Herfort, L., Damsté, J. S. S. and Schouten, S.: A novel proxy for terrestrial organic matter in sediments based on branched and isoprenoid tetraether lipids, Earth Planet. Sci. Lett., 224(1), 107–116, doi:https://doi.org/10.1016/j.epsl.2004.05.012, 2004.

Lapointe, L. E., Talbot, J., Fortier, D., Fréchette, B., Strauss, J., Kanevskiy, M. and Shur, Y.: Middle to late Wisconsinan climate and ecological changes in northern Alaska: Evidences from the Itkillik River Yedoma, Palaeogeogr. Palaeoclimatol. Palaeoecol., 485, 906–916, doi:https://doi.org/10.1016/j.palaeo.2017.08.006, 2017.

Lenz, J., Grosse, G., Jones, B. M., Walter Anthony, K. M., Bobrov, A., Wulf, S. and Wetterich, S.: Mid-Wisconsin to Holocene Permafrost and Landscape Dynamics based on a Drained Lake Basin Core from the Northern Seward Peninsula, Northwest Alaska, Permafr. Periglac. Process., 27(1), 56–75, doi:10.1002/ppp.1848, 2016.

Lenz, J., Jones, B. M., Fuchs, M. and Grosse, G.: C/N and d13C data from short sediment cores from 9 lakes in Western Alaska, [online] Available from: https://doi.pangaea.de/10.1594/PANGAEA.887848, 2018.

Peterse, F., Meer, J. van der, Schouten, S., Weijers, J. W. H., Fierer, N., Jackson, R. B., Kim, J.-H. and Damsté, J. S. S.: Revised calibration of the MBT–CBT paleotemperature proxy based on branched tetraether membrane lipids in surface soils, Geochim. Cosmochim. Acta, 96, 215–229, doi:https://doi.org/10.1016/j.gca.2012.08.011, 2012.

Peterse, F., Vonk, J. E., Holmes, R. M., Giosan, L., Zimov, N. and Eglinton, T. I.: Branched glycerol dialkyl glycerol tetraethers in Arctic lake sediments: Sources and implications for paleothermometry at high latitudes, J. Geophys. Res. Biogeosciences, 119(8), 1738–1754, doi:10.1002/2014JG002639, 2014.

Strauss, J., Schirrmeister, L., Wetterich, S., Borchers, A. and Davydov, S. P.: Grain-size properties and organic-carbon stock of Yedoma Ice Complex permafrost from the Kolyma lowland, northeastern Siberia, Glob. Biogeochem. Cycles, 26(3), doi:10.1029/2011GB004104, 2012.

Weijers, J. W.: Soil-derived branched tetraether membrane lipids in marine sediments: reconstruction of past continental climate and soil organic matter fluxes to the ocean, UU Dept. of Earth Sciences., 2007.